Scientific Report

# FET family fusion oncoproteins target the SWI/SNF chromatin remodeling complex

Malin Lindén[1],[†] (ID), Christer Thomsen[1],[2],[†], Pernilla Grundevik[1], Emma Jonasson[1], Daniel Andersson[1], Rikard Runnberg[1], Soheila Dolatabadi[1], Christoffer Vannas[1],[3], Manuel Luna Santamaría[1], Henrik Fagman[1],[2], Anders Ståhlberg[1],[2],[4],[*] (ID) & Pierre Åman[1],[2],[**] (ID)

## Abstract

Members of the human FET family of RNA-binding proteins, comprising FUS, EWSR1, and TAF15, are ubiquitously expressed and engage at several levels of gene regulation. Many sarcomas and leukemias are characterized by the expression of fusion oncogenes with FET genes as 5′ partners and alternative transcription factor-coding genes as 3′ partners. Here, we report that the N terminus of normal FET proteins and their oncogenic fusion counterparts interact with the SWI/SNF chromatin remodeling complex. In contrast to normal FET proteins, increased fractions of FET oncoproteins bind SWI/SNF, indicating a deregulated and enhanced interaction in cancer. Forced expression of FET oncogenes caused changes of global H3K27 trimethylation levels, accompanied by altered gene expression patterns suggesting a shift in the antagonistic balance between SWI/SNF and repressive polycomb group complexes. Thus, deregulation of SWI/SNF activity could provide a unifying pathogenic mechanism for the large group of tumors caused by FET fusion oncoproteins. These results may help to develop common strategies for therapy.

**Keywords** EWSR1-FLI1; FET proteins; FUS-DDIT3; fusion oncogenes; SWI/SNF chromatin remodeling complex

**Subject Categories** Cancer; Chromatin, Epigenetics, Genomics & Functional Genomics; Transcription

## Introduction

The FET family genes *FUS, EWSR1,* and *TAF15* (also known as *TLS, EWS,* and *TAF2N*, respectively) encode RNA-binding proteins (Fig 1A) that are proposed to link transcription with the subsequent steps of RNA splicing, processing and transport [1–5], localized translation [6], and micro-RNA processing [7]. Fusion oncogenes with FET genes as 5′ partners and alternative transcription factor-coding genes as 3′ partners (Fig 1B) are pathognomonic of many types of sarcoma and leukemia [8,9]. FET-oncogene-caused tumors contain few other mutations, indicating that FET fusion oncoproteins impact on crucial mechanisms in tumor development [9–15].

FET fusion oncoproteins invariably contain the N-terminal domains (NTDs) of the FET partners juxtaposed to DNA-binding parts of the transcription factor partners (Fig 1A and B). They are reported to act as aberrant transcription factors with the NTDs as strong trans-activator domains [16–25]. Forced expression or silencing of FET oncogenes affects tumor morphology and regulation of large numbers of genes, and changes the epigenetic landscape [24–27].

The three normal FET proteins contain central RNA recognition motifs (RRM) flanked by RGG repeat regions and potential single-stranded DNA- or RNA-binding zinc finger domains (Fig 1A) [28–30]. The NTDs largely consist of structurally disordered, prion-like degenerated SYGQ-rich repeats, and their compositions suggest functions in protein–protein interactions [31]. Their similarities are further underscored by the fact that they functionally replace each other as N-terminal partners in some FET fusion oncoproteins and tumor entities [28,32]. Given these observations, we hypothesized that the three FET-NTDs could act by binding the same key interaction partner. However, even with several FET-binding proteins identified, no such interaction partners have been reported and the role of the NTDs remains enigmatic. The aim of this study was to identify major interaction partners shared by the three FET-NTDs that might give clues to a common pathogenetic mechanism.

## Results and Discussion

### The FET-NTDs mediate binding of both normal and oncogenic FET proteins to SWI/SNF

We used bacterially expressed GST-tagged recombinant constructs in pulldown experiments with cell extracts for an unbiased analysis

1 Department of Pathology and Genetics, Sahlgrenska Cancer Center, Institute of Biomedicine, Sahlgrenska Academy, University of Gothenburg, Gothenburg, Sweden
2 Department of Clinical Pathology and Genetics, Sahlgrenska University Hospital, Gothenburg, Sweden
3 Department of Oncology, Sahlgrenska University Hospital, Gothenburg, Sweden
4 Wallenberg Centre for Molecular and Translational Medicine, University of Gothenburg, Gothenburg, Sweden
*Corresponding author. Tel: +46 317866735; E-mail: anders.stahlberg@gu.se
**Corresponding author. Tel: +46 317866732; E-mail: pierre.aman@gu.se
†These authors contributed equally to this work

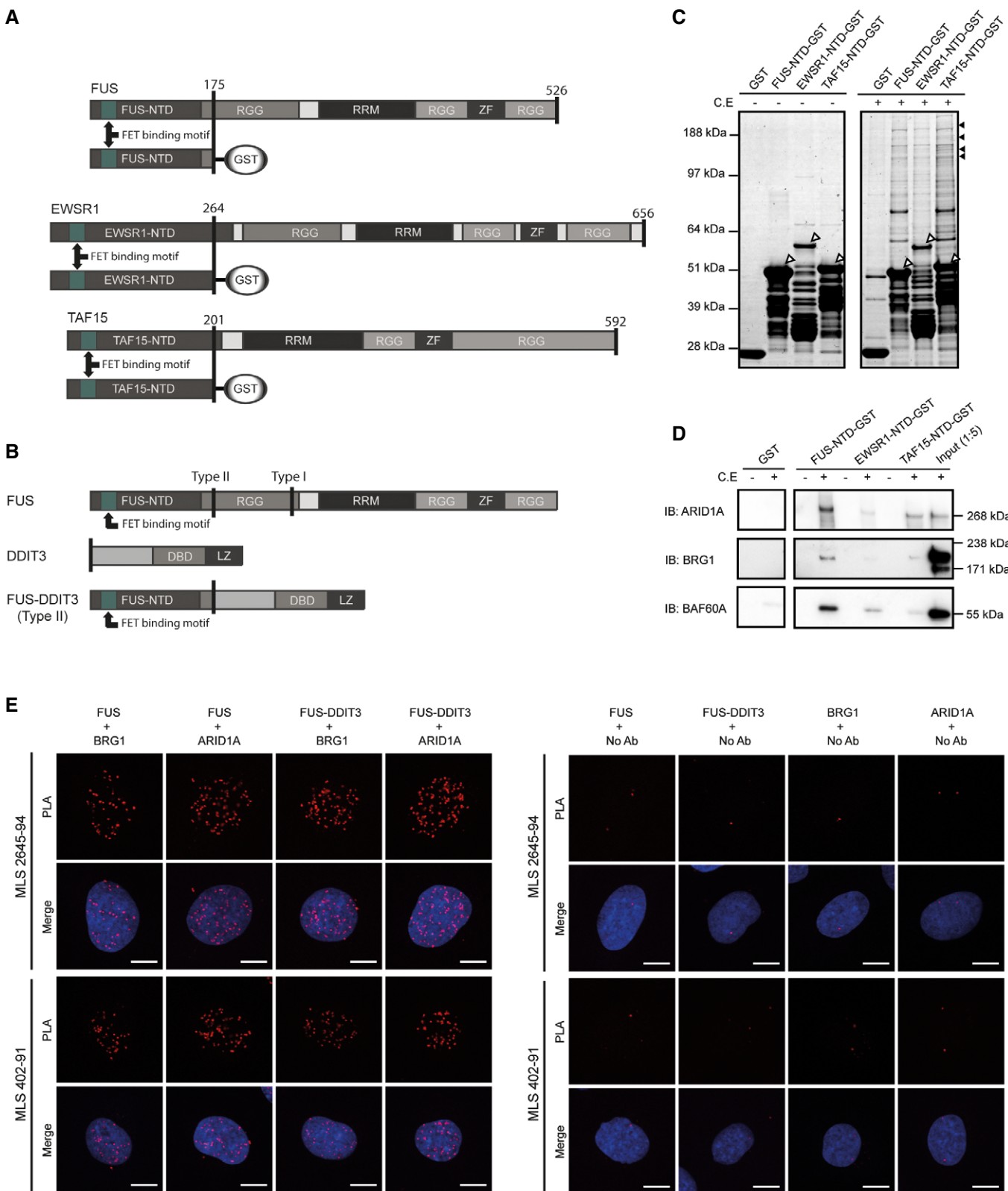

**Figure 1.**

of enrichment of FET-NTD-binding proteins (Fig 1A). SDS–PAGE analysis and protein staining revealed several high-molecular-weight proteins captured by all three FET-NTDs (Fig 1C). Mass

spectrometry analysis (MS) of excised gel bands identified peptides from core components of the SWI/SNF chromatin remodeling complex: ARID1A (BAF250A), BRG1 (SMARCA4), BAF170

**Figure 1.  Identification of proteins bound to recombinant FET-NTD baits.**

A   The three normal RNA-binding FET proteins consist of N-terminal repetitive and structurally disordered domains (NTD: N-terminal domain), central RNA-binding domains (RRM: RNA recognition motifs), zinc finger domains (ZF), and degenerated repeat regions (RGG: RGG repeat regions). The GST-tagged FET-NTD baits shown were used for pulldown experiments and represent the shortest parts commonly present in the FET fusion oncoproteins. The FET-binding motif is a conserved sequence required for complex formation between the three FET proteins. Amino acid numbers are indicated.

B   Schematic illustration of a representative FET fusion protein (FUS-DDIT3 type II) and its parental proteins. DBD: DNA-binding domain, LZ: leucine zipper domain. Type I and type II show locations of the two most common MLS fusion breakpoints in FUS.

C   Coomassie staining of SDS–PAGE-separated pulldown samples with/without cell extracts (C.E) using GST-tagged FET-NTDs as baits. Sepharose with bound GST was included as control. Background from recombinant protein baits (shown by white arrowheads) as well as smaller partial recombinant products is visualized in the left panel (C.E −). Several high-molecular-weight proteins are retained by the FET-NTDs (C.E +). Black arrowheads indicate protein bands and gel parts analyzed by mass spectrometry (see also Table EV1).

D   Immunoblot analysis (IB) of pulldown samples with FET-NTD baits with/without cell extracts (C.E). Antibodies against ARID1A, BRG1, and BAF60A were used for detection of SWI/SNF components. Input samples diluted 1:5 were included as a control.

E   *In situ* proximity ligation assays (PLA) using antibodies against BRG1, ARID1A, DDIT3, and C-terminal parts of normal FUS show protein complexes containing FUS/BRG1, FUS/ARID1A, FUS-DDIT3/BRG1, and FUS-DDIT3/ARID1A as red fluorescent spots in nuclei of MLS cell lines 2645-94 and 402-91. C-terminal parts of FUS are not present in the FUS-DDIT3 fusion protein, and normal DDIT3 is not expressed in these cell lines. Merged images also include DAPI nuclear counterstain in blue. Combinations of primary antibodies are used to detect interactions (left panel). In control experiments (right panel), one primary antibody is omitted to evaluate the background fluorescent signals. Scale bars = 10 μm.

Source data are available online for this figure.

(SMARCC2), and BAF155 (SMARCC1) (Table EV1). The enrichment of SWI/SNF core components was also confirmed by immunoblot analysis (Fig 1D). Taken together, the results show that all three FET-NTDs have the capacity to bind the SWI/SNF chromatin remodeling complex. This large multi-subunit complex consists of around 15 tightly bound core proteins that control gene expression by ATP-driven repositioning of nucleosomes and eviction of repressing polycomb complexes [33,34]. Incorporation of alternative core proteins from closely related genes results in many variant SWI/SNF complexes. The importance of SWI/SNF activities in cells and tissues is evident from knock-out experiments and by mutations affecting SWI/SNF components in many forms of cancer [35].

Normal FET proteins shuttle between nucleus and cytoplasm, whereas the fusion oncoproteins are mainly restricted to the nuclei [5,36,37]. *In situ* proximity ligation assays visualized nuclear complexes containing normal FUS or oncogenic FUS-DDIT3 bound to SWI/SNF core proteins BRG1 and ARID1A in myxoid liposarcoma (MLS) tumor cells (Fig 1E). The results confirm the binding of normal FUS and FUS-DDIT3 oncoprotein to SWI/SNF *in situ* and show a nuclear localization of these complexes.

## Interaction with SWI/SNF is robust and mediated by conserved motifs in FUS-NTD

For further analysis of the FET-NTD binding properties, we selected the FUS-NTD since it is the shortest oncogenic NTD variant, differing from EWSR1 and TAF15 NTDs mainly in the numbers of degenerated SYGQ-rich repeats [28,31,38]. To define the FET-NTD sequences that bind SWI/SNF, we constructed a series of FUS-NTD deletion mutant baits (Fig 2A). The binding to SWI/SNF core proteins failed when amino acids 31–66 were deleted (Fig 2B). This sequence contains a conserved 26 amino acid "FET binding motif", also important for formation of homo- or hetero-complexes between the three normal FET proteins and the cytoplasmic protein plectin (Fig 1A) [31,39]. Neither DNase nor RNase treatment interrupted the binding of FUS-NTD to SWI/SNF, excluding nucleic acids as mediators of the interaction (Fig 2C). Taken together, these results show that the binding of FET proteins to SWI/SNF is independent of nucleic acids and that only a minor part of the repetitive domain

may be sufficient for the binding. The results are compatible with the FET proteins binding SWI/SNF either as single molecules, or as multimeric FET complexes including fusion oncoproteins.

SWI/SNF binding to the recombinant FUS-NTD was further tested by stringency washing with increasing NaCl concentrations. Immunoblot analysis of eluates showed loss of the BRG1 component at more than 250 mM NaCl, while ARID1A, BAF170, BAF155, BAF60A, BAF47, SS18, FUS, and EWSR1 remained bound to the recombinant FUS-NTD at 1 M salt (Fig 2D). This indicates a very robust FET-NTD binding to several SWI/SNF core proteins but a weaker, perhaps indirect binding of BRG1 through other SWI/SNF components.

## Major fractions of FET oncoproteins interact with SWI/SNF and show no binding competition with normal FET proteins

To further confirm the SWI/SNF binding of normal and oncogenic FET proteins, we used a BRG1-specific antibody for immunoprecipitation (IP) of SWI/SNF complexes from MLS and Ewing sarcoma (EWS) cell lines carrying four different FET oncogenes (*FUS-DDIT3* type I, *FUS-DDIT3* type II, *EWSR1-FLI1,* and *EWSR1-ERG*) (Table 1, Fig EV1A and B and Table EV2). MS and immunoblot analysis of immunoprecipitates demonstrated presence of normal FUS and EWSR1, as well as the oncogenic FUS-DDIT3 and EWSR1-FLI1/ERG, confirming that normal and oncogenic FET proteins bind the SWI/SNF complex. The analysis also showed that these tumor cell lines produced complete sets of SWI/SNF core proteins (Table 1 and Table EV2).

We then used anti-BRG1 IP to quantify the interaction between the FET proteins and SWI/SNF in cell lines expressing FUS-DDIT3 and EWSR1-FLI1 (Fig 3A and B). Normal FET proteins and SWI/SNF complexes are abundant in most tissues and cell types and with our results that FET-NTDs bind SWI/SNF, extensive interactions between them could be expected. However, only minor fractions of normal FET proteins bound to precipitated SWI/SNF, suggesting that the normal FET protein binding is regulated and perhaps confined to restricted chromatin locations, activities or SWI/SNF complex subtypes. In contrast, FET oncoproteins are weakly expressed [40], but major fractions were bound to

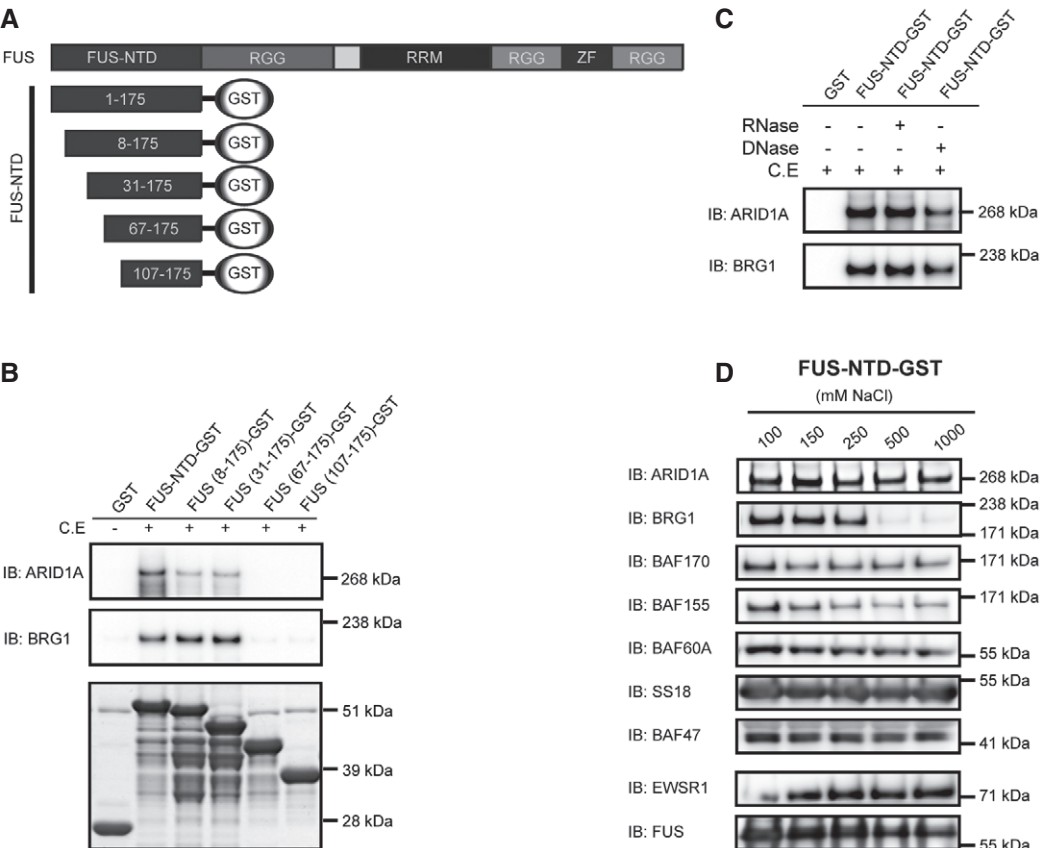

**Figure 2. Analysis of the binding between FUS-NTD and SWI/SNF core components.**

A  Schematic illustration of FUS-NTD-GST truncation constructs. Amino acid numbers are indicated.
B  Immunoblot analysis (IB) of GST-pulldown samples with deletion mutants of FUS-NTD-GST as baits with/without cell extract (C.E). ARID1A and BRG1 were used as tracers of SWI/SNF binding. Coomassie staining of recombinant proteins is shown in the lower panel. Amino acid numbers are indicated.
C  Immunoblot analysis (IB) of FUS-NTD-GST-pulldown samples of cell extracts (C.E.) after treatment with RNase or DNase, using antibodies against ARID1A and BRG1.
D  Immunoblot analysis (IB) of SWI/SNF components (ARID1A, BRG1, BAF170, BAF155, BAF60A, SS18, and BAF47) and full-length FET proteins (EWSR1 and FUS) remaining on sepharose-bound FUS-NTD-GST-pulldown baits after increasing stringency washes with 100, 150, 250, 500, or 1,000 mM NaCl. Note loss of BRG1 at NaCl concentration > 250 mM.

Source data are available online for this figure.

BRG1-precipitated complexes indicating a dysregulated interaction with SWI/SNF. Furthermore, gentle formic acid elution released substantial amounts of EWSR1 whereas elution with the strong LDS detergent was needed to release the FUS-DDIT3 fusion protein from anti-BRG1-captured complexes (Fig EV1A and B). Taken together, these results demonstrate that compared to normal FET proteins, a larger fraction of FET oncoproteins bind, with an increased binding strength, to SWI/SNF.

Binding of normal and oncogenic FET proteins to SWI/SNF complexes could be expected to result in a competition for binding sites. To test this hypothesis, we expressed DsRED- or EGFP-tagged FUS-DDIT3, EWSR1-FLI1 or the tags alone in human HT1080 fibrosarcoma cells and quantified the normal and oncogenic FET proteins that co-precipitated with BRG1. Although the results show massive binding of the FET oncoproteins to BRG1 precipitates, no reduction of BRG1-bound FUS or EWSR1 proteins was observed (Fig 3C and D).

Based on our combined results, we propose alternative models for the binding between FET fusion oncoproteins and SWI/SNF

chromatin remodeling complexes (Fig 3E). The divergent IP-binding/elution patterns between the normal and oncogenic FET proteins suggest that the fusion oncoproteins bind directly to SWI/SNF and not indirectly by multimerization with normal FET proteins (Fig 3E, models I and II). The lack of competition also suggests that normal and oncogenic FET proteins bind different variants of SWI/SNF or to different sites on the SWI/SNF complex (Fig 3E, models III and IV). As mentioned earlier, FET oncoproteins binds directly to normal FET proteins [31], and normal FET proteins form homo- and hetero-complexes. Furthermore, the many co-existing variants of SWI/SNF complexes [35,41] may have divergent binding properties. Several alternative complexes between FET proteins and SWI/SNF variants may thus co-exist in the tumor cells.

**SWI/SNF complexes bound by FET oncoproteins show normal core protein composition**

The composition of the SWI/SNF complex is changed by the SS18-SSX fusion oncoprotein in synovial sarcomas. The fusion protein

**Table 1.** Summary of SWI/SNF components as well as FET proteins (wild-type and fusion oncoproteins) in the bound fraction after BRG1 Co-IP of MLS 402-91, MLS 2645-94, EWS TC-71, and EWS IOR/CAR, after DDIT3 Co-IP of MLS 402-91 and after FLI1 Co-IP of EWS TC-71 as verified by mass spectrometry (MS), and by immunoblot (IB) for key proteins. Note that some SWI/SNF components are represented by one of several alternative variants, for example BAF45A-D. Lists of peptides for each mass spectrometry hit are shown in Tables EV2 and EV3.

| Protein | Alternative name | BRG1 Co-IP | | | | DDIT3 Co-IP | FLI1 Co-IP |
| --- | --- | --- | --- | --- | --- | --- | --- |
| | | MLS 402-91 | MLS 2645-94 | EWS TC-71 | EWS IOR/CAR | MLS 402-91 | EWS TC-71 |
| BRG1 | SMARCA4 | MS, IB | MS, IB | MS, IB | MS, IB | MS, IB | MS, IB |
| BRM | SMARCA2 | - | - | - | - | (MS) | MS |
| ARID1A | BAF250A | MS | MS | MS | MS | MS | MS |
| ARID1B | BAF250B | - | MS | - | - | MS | MS |
| ARID2 | BAF200 | - | - | MS | MS | MS | MS |
| BAF170 | SMARCC2 | MS | MS | MS | MS | MS | MS |
| BAF155 | SMARCC1 | MS | MS | MS | MS | MS | MS |
| BAF60A | SMARCD1 | MS | MS | MS | MS | MS | MS |
| BAF60B | SMARCD2 | MS | MS | MS | MS | MS | MS |
| BAF60C | SMARCD3 | MS | - | MS | MS | (MS) | MS |
| BAF57 | SMARCE1 | MS | MS | MS | MS | MS | MS |
| BAF47 | SMARCB1 | MS | MS | MS | MS | MS | MS, IB |
| BAF53A | ACTL6A | MS | MS | MS | MS | MS | MS |
| BAF53B | ACTL6B | - | - | - | - | - | - |
| BAF45A | PHF10 | - | - | MS | MS | - | MS |
| BAF45B | DPF1 | - | - | - | - | - | - |
| BAF45C | DPF3 | - | - | - | - | MS | - |
| BAF45D | DPF2 | MS | MS | MS | MS | MS | MS |
| BAF180 | PBRM1 | MS | MS | MS | MS | MS | MS |
| SS18 | SYT | -, IB | MS | MS | MS | MS | -, IB |
| FUS | TLS | MS | MS | MS | MS | MS | MS |
| EWSR1 | EWS | MS, IB | MS, IB | MS, IB | MS, IB | MS | MS |
| TAF15 | TAF2N | - | MS | MS | MS | MS | MS |
| FUS-DDIT3 | TLS-CHOP | -, IB | -, IB | | | MS, IB | |
| EWSR1-FLI1 | | | | -, IB | | | MS, IB |
| EWSR1-ERG | | | | | MS | | |

"-": no tryptic peptides detected, " ": not applicable, "()": no unique peptides detected.

containing N-terminal parts of the SS18 SWI/SNF core protein evicts the BAF47 protein from the complex [42,43]. Even though FET proteins or their fusion partners are not considered core SWI/SNF constituents [35], they could theoretically cause a change in SWI/SNF composition. To test this possibility, we precipitated FUS-DDIT3 and EWSR1-FLI1 oncoproteins from MLS and EWS cell lines using DDIT3- and FLI1-specific antibodies. Since no normal DDIT3 or FLI1 is expressed in these cell lines, the antibodies only precipitate the fusion oncoproteins and their interacting proteins. MS analysis of the precipitates demonstrated complete sets of SWI/SNF core proteins co-precipitated with the fusion oncoproteins (Table 1 and Table EV3). These results rule out the possibility that oncogenic FET proteins cause loss of core proteins from SWI/SNF and indicate a different mechanism of action compared to the SS18-SSX fusion oncoprotein [42,43]. It also indicates that the FET oncoproteins bind several SWI/SNF variants as all the alternative SWI/SNF core proteins found in the cells were also precipitated with the FET oncoproteins.

## Forced expression of FET oncoproteins causes increased H3K27me3 levels

The binding of FET oncoproteins to SWI/SNF prompted us to investigate whether this could lead to any functional effects. SWI/SNF has been reported to balance the suppressive H3K27 trimethylation (H3K27me3) caused by the PRC2 polycomb repressor complex [33,34]. Oncogenic mutations of SWI/SNF components have been reported to disturb this balance, leading to increased polycomb activity and H3K27me3 levels [34]. Elevated H3K27me3 was for example reported in cells lacking the SWI/SNF core BAF47/SMARCB1 protein [44]. Our immunoblot analysis of histone modifications in protein extracts from human HT1080 sarcoma cells

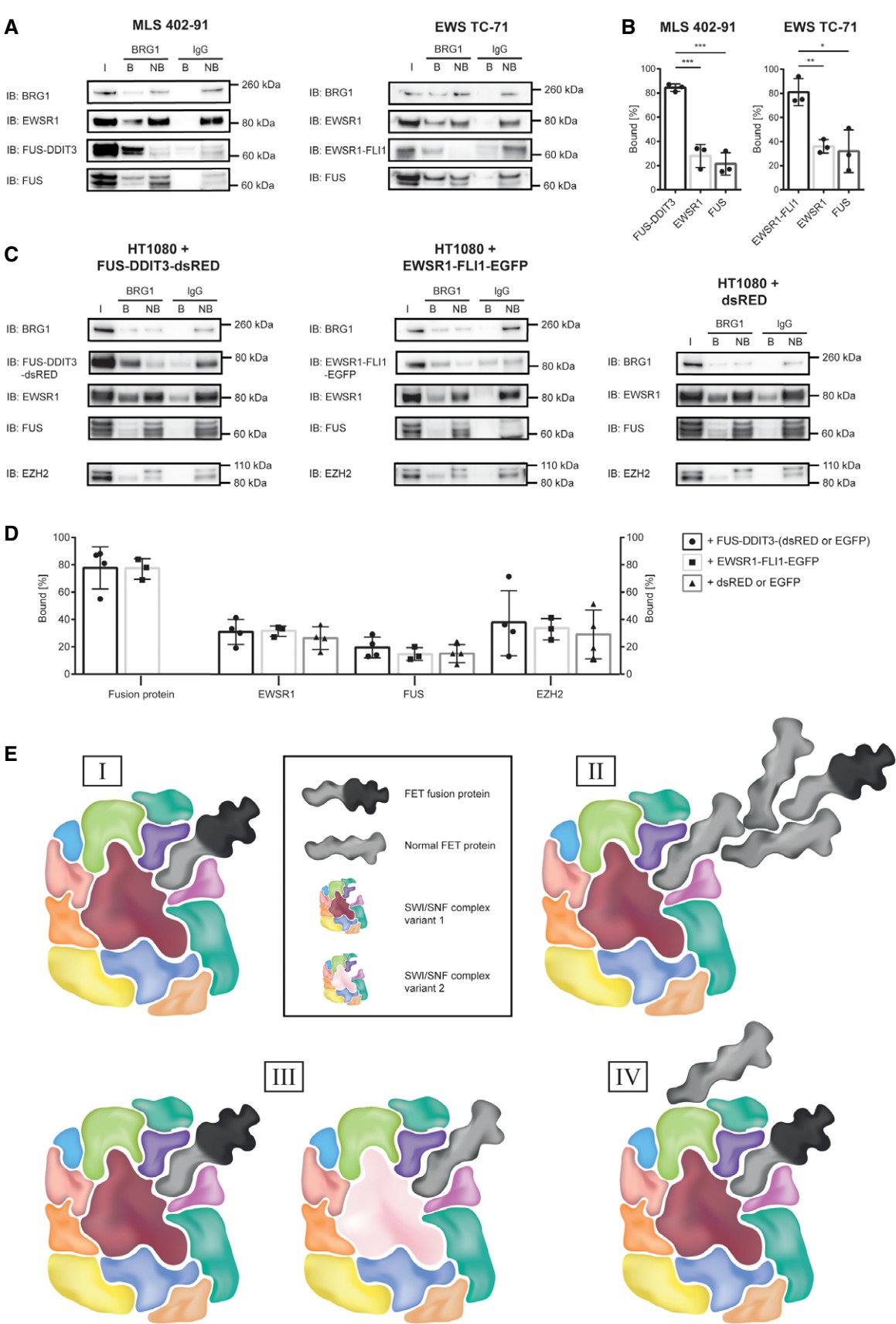

Figure 3.

◀

**Figure 3. Co-immunoprecipitation of SWI/SNF and FET proteins in sarcoma cell lines.**

A　Immunoblot analysis (IB) of proteins co-immunoprecipitated with nuclear extracted SWI/SNF from MLS cell line 402-91 and EWS cell line TC-71. Antibodies against BRG1, DDIT3 (FUS-DDIT3), the C-terminal parts of normal FUS and EWSR1, and FLI1 (EWSR1-FLI1) were used for detection. In order to directly quantify the fraction of bound and non-bound protein, relative amounts of protein for each IP-sample were loaded on the gel, with consideration taken for dilutions during the immunoprecipitation procedure. I: input of nuclear extract, B: bound proteins, NB: proteins not bound. One representative immunoblot is shown. Immunoblots from all replicates are shown in the source data.

B　Graphs showing the percentage of bound to total (bound + non-bound) signal intensities from immunoblots for FUS-DDIT3, EWSR1, and FUS in MLS 402-91 cell line and EWSR1-FLI1, EWSR1, and FUS in EWS TC-71 cell line. Mean $\pm$ SEM is shown with individual replicates indicated by circles ($n = 3$). Student's $t$-test, *$P < 0.05$, **$P < 0.01$, ***$P < 0.001$. Original data for all quantifications, including $P$-values, are shown as source data.

C　Immunoblot analysis of proteins co-immunoprecipitated with nuclear extracted SWI/SNF from the model cell line HT1080 with overexpressed FUS-DDIT3-dsRED, EWSR1-FLI1-EGFP, or dsRED is shown as representative immunoblots. Antibodies against BRG1, DDIT3 (FUS-DDIT3-dsRED), the C-terminal parts of normal FUS and EWSR1, EZH2, and FLI1 (EWSR1-FLI1-EGFP) were used for detection. In order to directly quantify the fraction of bound and non-bound protein, relative amounts of protein for each IP-sample were loaded on the gel, with consideration taken for dilutions during the immunoprecipitation procedure. I: input of nuclear extract, B: bound proteins, NB: proteins not bound. Immunoblots from all replicates are shown in the source data (with either dsRED- or EGFP-tagged proteins expressed).

D　Graphs showing the percentage of bound to total (bound + non-bound) signal intensities from immunoblots for the fusion protein (FUS-DDIT3 or EWSR1-FLI1), EWSR1, FUS, and EZH2 in the model cell line HT1080 with overexpressed FUS-DDIT3-dsRED/-EGFP ($n = 4$, circles), EWSR1-FLI1-EGFP ($n = 3$, squares) or dsRED/EGFP ($n = 4$, triangles). Mean $\pm$ SEM is shown with individual replicates indicated. Student's $t$-test, no significant changes, $P > 0.05$. Original data for all quantifications, including $P$-values, are shown as source data.

E　Four alternative models for binding of normal and oncogenic FET proteins to SWI/SNF. (I): FET oncoproteins bind directly to SWI/SNF. (II): normal FET proteins form homo- and hetero-complexes and mediate binding of FET oncoproteins. (III): FET oncoproteins and normal FET proteins bind to different variants of the SWI/SNF complex with distinct biochemical compositions. (IV): FET oncoproteins and normal FET proteins bind to different sites on the SWI/SNF complex.

Source data are available online for this figure.

showed that forced expression of the oncogene *FUS-DDIT3* leads to significantly higher H3K27me3 levels (Fig 4A and B, and complete data set in Fig EV2A–I). *EWSR1-FLI1* transfections also lead to increased but more variable H3K27me3 levels. Additional experiments verified the fusion oncoprotein-induced trimethylation of H3K27 (Fig EV2C, D, H and I). Treatment of the cells with the EZH2 inhibitor tazemetostat dramatically reduced the H3K27me3 level proving that the quantitative Western blot assay was functional (Fig EV2E and J). H3K4me3 levels were largely unaffected, and effects on H3K27 acetylation varied but showed no consistent trend. We observed a small, not statistically significant, increase in EZH2 expression (Fig 4A and B) while no increased co-precipitation with BRG1 was seen (Fig 3C and D). A plausible explanation for these results is that binding of FET oncoprotein to SWI/SNF negatively impacts its PRC2-balancing function leading to increased PRC2/EZH2 activity. The observed effects of FET oncogenes are thus similar to those caused by some oncogenic mutations in SWI/SNF core complex proteins. The increase in H3K27me3 levels may seem minor but could translate to changed regulation of hundreds of genes.

Our results are in parts at odds with recently published studies on EWS cell lines, reporting no effects on H3K27me3 levels but instead increased H3K27 acetylation levels at specific genomic sites [24,45]. The diverging results could be explained by the use of different experimental systems. The normal FLI1 protein is constitutively expressed in HT1080 cells used in our study (Fig EV2D) while this protein is absent in EWS cells [46]. The constitutive FLI1 expression may block EWSR1-FLI1 binding and the histone acetylation effects induced by this oncoprotein. Also, ChIP-seq-based assays used for the EWS-specific studies show genomic sites and distribution of histone modifications, but may fail to detect minor but wide spread changes in H3K27me3.

## FUS-DDIT3-altered gene expression patterns overlap with PRC2-regulated gene-sets

To further investigate downstream effects of the FET oncogenes *FUS-DDIT3* and *EWSR1-FLI1,* we performed RNA-seq to compare the

gene expression patterns of FET-oncogene transfected and wild-type HT1080 cells ($n = 3$–4) (Fig 4C and source data). Lists of > 2-fold regulated genes were compared with the database of gene-sets changed after "chemical and genetic perturbations" (CGP, 3,433 gene-sets) using the Gene Set Enrichments Analysis (GSEA) tool. FUS-DDIT3 downregulated genes overlapped significantly with gene-sets upregulated in cells after knock-out of the PRC2 component EZH2 ($q$-value $1.68 \times 10^{-31}$) or reconstitution of SWI/SNF component BAF57/SMARCE1 ($q$-value $2.37 \times 10^{-31}$). Deletion of NIPP1 (nuclear inhibitor of protein phosphatase-1) leads to degradation of EZH2 expression [47]. In line with this observation, FUS-DDIT3 downregulated genes overlapped significantly with genes upregulated after knockdown of NIPP1 ($q$-value $2.9 \times 10^{-22}$), further underscoring the effect on PRC2 activity. These results are compatible with our hypothesis that FUS-DDIT3-induced downregulation of genes involves disruption of the SWI/SNF-PRC2 balance (Fig 4C). In contrast, FUS-DDIT3 upregulated genes showed no overlap with any gene-set at comparably low significance.

EWSR1-FLI1 regulated genes showed significant overlaps with gene-sets from conditions where EWSR1-FLI1 expression was modified. Moreover, our upregulated genes overlapped with CGP gene-sets containing EWSR1-FLI1 upregulated gene-sets and our downregulated genes matched with CPG-sets containing EWSR1-FLI1 downregulated genes, indicating that the EWSR1-FLI1 transfected HT1080 cells mimicked EWS cells (Fig 4C and source data). In contrast to the FUS-DDIT3 expressing cells, there were no comparably low $q$-score overlaps between EWSR1-FLI regulated gene lists and gene-sets from PRC2- or SWI/SNF-regulated conditions. This indicates divergent effects of FUS-DDIT3 and EWSR1-FLI1 in HT1080 cells even though SWI/SNF is the major binding partner of both oncoproteins.

The divergent effects between FUS-DDIT3 and EWSR1-FLI1 may be explained by the very different properties of the DDIT3 and FLI1 transcription factor partners. DDIT3 is a leucine zipper-containing, dimer-forming transcription factor of the CEBP family. It has, however, an acidic DNA-binding domain and is therefore considered to exert a dominant negative function, blocking DNA binding of most

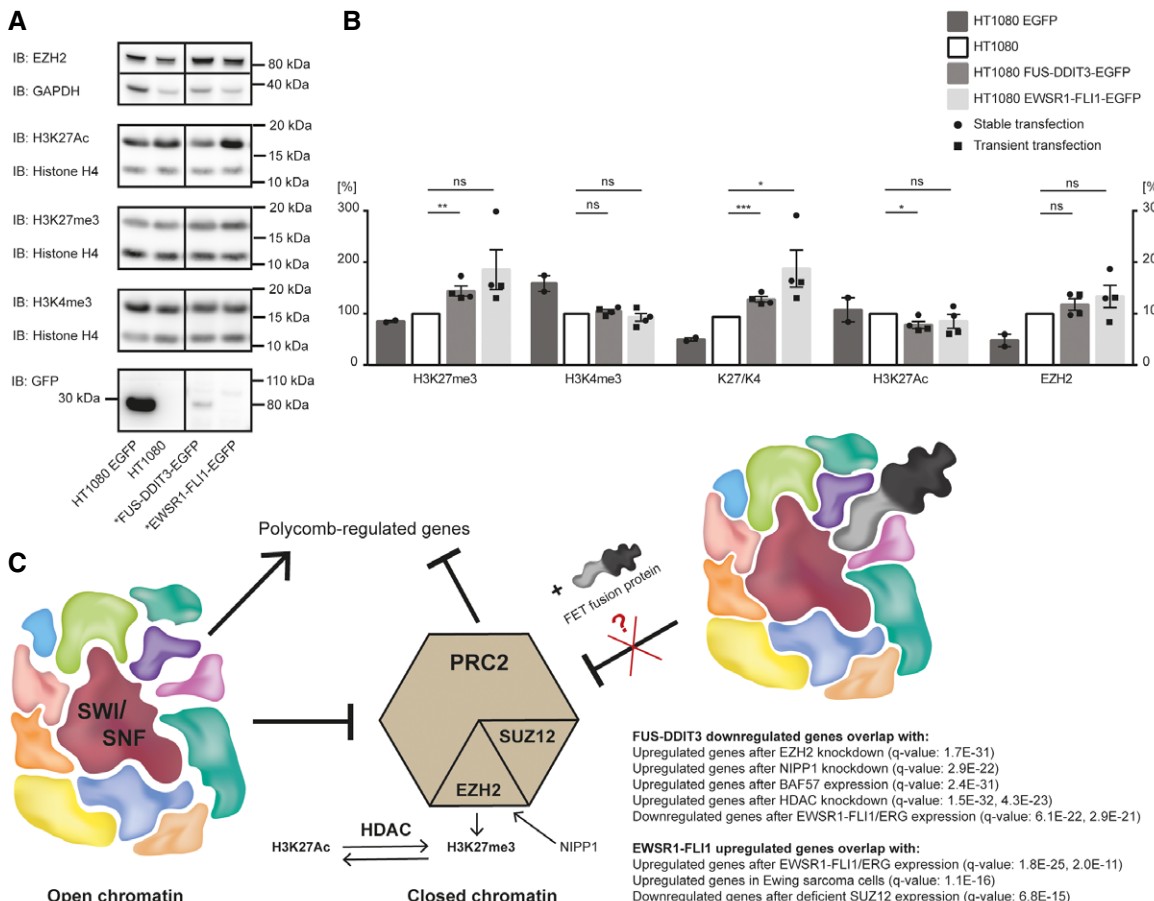

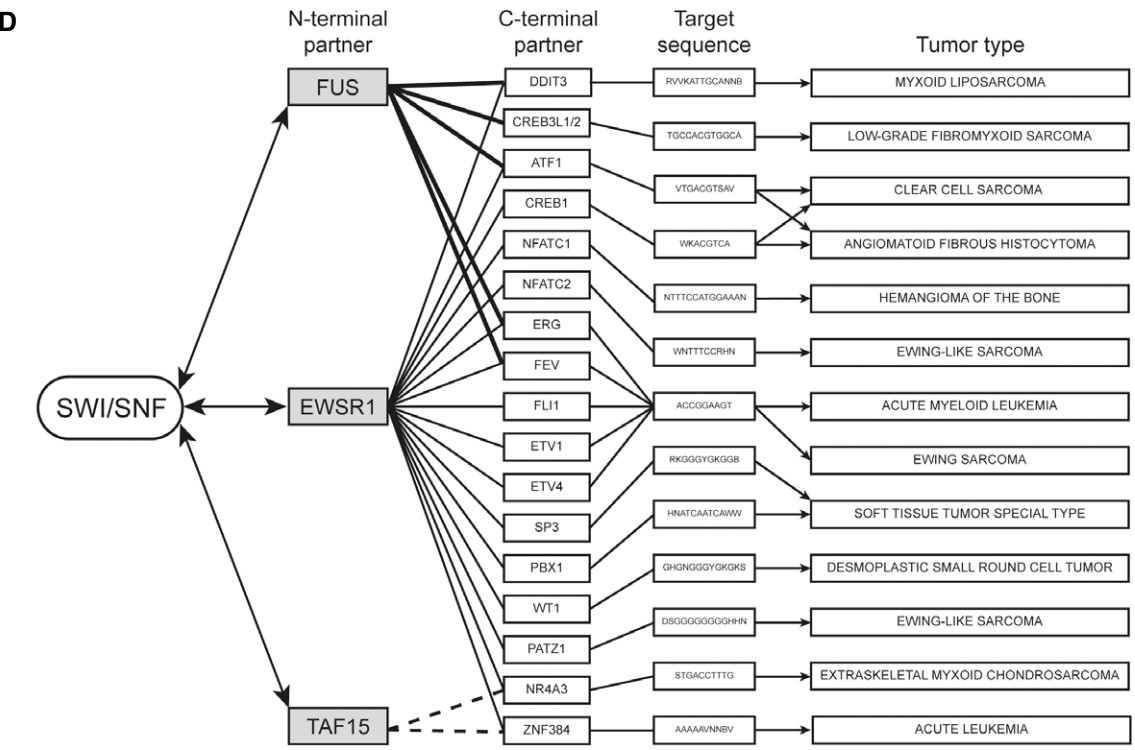

**Figure 4.**

◄

**Figure 4. FET-oncogene-induced changes in H3K27 trimethylation and downstream effects.**

A Immunoblot analysis (IB) of histone modifications in stably transfected HT1080 cell lines expressing EGFP, FUS-DDIT3-EGFP, or EWSR1-FLI1-EGFP. Antibodies against H3K27Ac, H3K27me3, H3K4me3, and histone loading control H4 for detection of histone modifications and antibodies against EZH2 and loading control GAPDH to evaluate catalytic PRC2 amount. Immunoblot analysis with GFP antibody is shown to verify expression of FET fusion oncoproteins and EGFP. One representative immunoblot is shown. More immunoblots including a second stable biological replicate and analysis of histone modifications after transient transfection (24 and 48 h) are shown in Fig EV2A–D.

B Graphs showing amount of protein (H3K27me3, H3K4me3, ratio H3K27me3/H3K4me3, H3K27Ac, and EZH2) quantified from immunoblots relative each corresponding loading control H4 or GAPDH, normalized to parental HT1080. Mean ± SEM is shown with individual replicates indicated by circles from two experiments with stable transfection (see Figs 4A, and EV2A and F) and by squares from two experiments with transient transfection (24 and 48 h, see Fig EV2B and G), $n = 4$. Student's $t$-test, ns = not significant, $*P < 0.05$, $**P < 0.01$, $***P < 0.001$. Original data for all quantifications, including $P$-values, are shown as source data.

C SWI/SNF opposes the polycomb complex PRC2. The catalytic subunit of PRC2, EZH2, catalyzes the trimethylation of Lys27 on histone H3 (H3K27me3), a chromatin modification associated with closed chromatin, leading to downregulation of polycomb-regulated genes. FET fusion binding to SWI/SNF might compromise the function of SWI/SNF, including polycomb opposition. In particular, genes downregulated when FUS-DDIT3 is overexpressed in HT1080 cells (RNA-seq data, $n = 3$, compared to control, $n = 4$) overlap with upregulated gene-sets after EZH2 knockdown, NIPP1 knockdown and HDAC knockdown, and BAF57 (SMARCE1) reconstitution in a SMARCE1 null cell line. RNA-seq data of HT1080 wt ($n = 4$), HT1080 EGFP ($n = 4$), HT1080 FUS-DDIT3-EGFP ($n = 3$), and HT1080 EWSR1-FLI1-EGFP ($n = 4$) were generated and genes > twofold regulated were analyzed using the molecular signature database (MSigDB). Gene lists were compared to the gene-set collection "chemical and genetic perturbations" and top 20 gene-sets for each comparison are shown in the source data. $q$-values (FDR-adjusted $P$-values) are indicated in parentheses.

D Schematic presentation of FET family of fusion oncoproteins, targeted DNA sequences, and associated tumors. Left: FET N-terminal fusion partners and binding to SWI/SNF. Center: C-terminal transcription factor fusion partners and their DNA target sequences (JASPAR database). Right: Tumor types caused by respective fusion oncogene. Note tumor type specificity for each fusion oncoprotein and that the FET-NTDs replace each other as fusion partners in some entities. Only a selection of the FET family fusion oncoproteins and tumors is shown as more members are continuously discovered.

Source data are available online for this figure.

dimerization partners [48]. In support of this view, ectopic expression of DDIT3 causes mainly negative regulation of affected genes [25,49]. ER stress, however, induces strong co-expression of DDIT3 and dimerization partner ATF4 that results in DNA-binding dimer pairs [50]. In MLS, where DDIT3 is ectopically expressed without ATF4, the number of DNA target sites is limited. In contrast, EWSR1-FII1 may bind tens of thousands potential DNA-binding sites in the genome [51,52], and the resulting SWI/SNF recruitment and activation of enhancers in these sites are considered an important part of the oncogenic mechanism. The normal FLI1 protein was reported to bind SWI/SNF itself and thus contribute to the FET oncoprotein binding. Here, we tested if normal DDIT3 could bind SWI/SNF. Experiments in HT1080 cells stably expressing GFP-tagged DDIT3 showed a weak co-precipitation of DDIT3 with BRG1 (Fig EV3A and B). These results indicate that both the FUS and DDIT3 parts contribute to the strong binding of FUS-DDIT3 to SWI/SNF. FUS-DDIT3 and EWSR1-FLI1 are thus highly divergent with regard to their DNA-binding profiles but share the SWI/SNF interaction. This indicates that the SWI/SNF binding by itself may be the important common oncogenic mechanism within the FET family oncogenes.

### A unifying pathogenic mechanism for the FET family of fusion oncogenes

The many variants of FET fusion oncoproteins are, with few exceptions, specific for one tumor type each (Fig 4D), and previous studies have shown that FET oncogenes are instructive for tumor morphology and gene expression patterns [25,53]. From Fig 4D, it is obvious that the transcription factor partners and their sequence specificity determine the tumor type. The recruitment of SWI/SNF to genomic sites determined by the FET oncoprotein was reported by Boulay *et al* [45]. Our results, showing that all FET-NTDs bind SWI/SNF, suggest that this aberrant SWI/SNF recruitment is a central mechanism behind the tumor type specificity and reflects a common pathogenic mechanism.

Malignant rhabdoid tumors and synovial sarcoma carry mutations in the BAF47/INI1/SNF5/SMARCB1 and SS18 SWI/SNF core components, respectively, that impair SWI/SNF functions. These tumors contain few additional mutations and are genetically stable [54,55], indicating the central role of SWI/SNF mutations in tumor development. Similarly, FET-oncogene-caused tumors are genetically stable with few other mutations and importantly, no reports of SWI/SNF mutations. Our data instead suggest that FET oncoproteins bind SWI/SNF and compromise its function. These common features of FET-oncogene-caused tumors, malignant rhabdoid tumors, and synovial sarcomas further underscore the importance of SWI/SNF and chromatin remodeling in cancer development and point to a novel unifying pathogenic mechanism for FET oncoprotein-associated tumors. Targeting of this mechanism may be a fruitful avenue for new therapies against all entities of this large group of tumors.

## Materials and Methods

### Cell culture

The Raji Burkitt lymphoma cell line [56] was a kind gift from Dr. Georg Klein, Karolinska Institutet. Myxoid liposarcoma (MLS) cell lines 402-91 and 2645-94 were established by us from MLS tumor tissues [57]. The fibrosarcoma cell line HT1080 [58] was obtained from ATCC (CCL-221, Manassas, VA, USA) and stable clones expressing FUS-DDIT3-EGFP, EWSR1-FLI1-EGFP, DDIT3-EGFP, and EGFP were established as described [59]. Possible mutations in genes encoding SWI/SNF components were ruled out by inspection of the COSMIC database and our own analysis on protein level. All cell lines were routinely screened for mycoplasma infections and, except for EWS cell lines, cultured in RPMI1640 with GlutaMAX. The EWS cell lines TC-71 and IOR/CAR were kind gifts from Dr. Katia Scotlandi, University of Bologna, and cultured in IMDM GlutaMAX. Culture media were supplemented with 5 or 10% fetal bovine serum, 100 U/ml penicillin, and 100 µg/ml streptomycin.

Stable expression of EGFP constructs were maintained by addition of 500 µg/ml Geneticin. All media and supplements were obtained from Life Technologies (Carlsbad, CA, USA). Cells were maintained at 37°C with air containing 5% $CO_2$. The unique fusion oncogene content of all used sarcoma cell lines was confirmed by RT–PCR analysis (Fig EV4).

An expression vector containing the full coding region of EWSR1-FLI1 (type 1) in pEGFP-N1 (Clontech, Mountain View, CA, USA) was made using the primers EWSR1XhoIF: ATACTCGAGATGGCGTC CACGGATTACAGTACC and FLI1Sal1R: ATAGTCGACCCGTAGTA GCTGCCTAAGTGTGAAGG in the same way as described for FUS-DDIT3 [59]. Transient transfection of HT1080 cells with the pEGFP-N1 expression vector (empty or containing the fusion oncogenes FUS-DDIT3 or EWSR1-FLI1) or with pDsRED1-N1 (Clontech) expression vector (empty or with FUS-DDIT3, cloned same way as in pEGFP-N1) was done using FuGENE® 6 Transfection Reagent (Promega, Madison, WI, USA) according to the manufacturer's recommendations. Cells were transfected the day after seeding, at 60% confluency with a transfection reagent (µl) to DNA (µg) ratio of 3:1. Nuclear extracts were made 24 h after transfection for dsRED/EGFP constructs, and whole-cell extracts were made 24 and 48 h after transfection for EGFP constructs to study histone modifications.

For PLA analysis, cells were grown on collagen I coated 8-well culture slides (BD Biosciences, San Jose, CA, USA). For tazemetostat treatment (EZH2-inhibition), cells were seeded on 6-wells plates and treated with 5 µM tazemetostat (EPZ-6438, Selleckchem, Munich, Germany) dissolved in DMSO, or DMSO-control, for 72 h before whole-cell extraction.

**Recombinant protein expression and pulldown**

Vectors encoding GST fusion proteins, recombinant protein expression, and purification were previously described [39]. Briefly, expression vectors were transformed to Rosetta DE3 pLysS (Novagen, Merck, Darmstadt, Germany) and inoculated in Luria broth (MP Biomedicals, Santa Ana, California, USA) with 50 µg/ml ampicillin and 34 µg/ml chloramphenicol (both Sigma-Aldrich, St. Louis, MO, USA). Bacteria were grown overnight at 37°C with orbital shaking, diluted 1:20, and grown to an $OD_{600}$ of 0.6 after which protein expression was induced by addition of IPTG (Merck) to a final concentration of 1 mM. After 4 h, bacteria were harvested by centrifugation at 6,000 $g$ for 10 min at 4°C, frozen in liquid $N_2$, and stored at −80°C. Pellets were thawed and resuspended in ice-cold lysis buffer (50 mM $NaH_2PO_4$ pH 7.5, 0.5% NP-40, 300 mM NaCl) supplemented with 5 mM DTT and protease inhibitors (Roche Diagnostics, Mannheim, Germany) followed by sonication. Samples were centrifuged at 12,000 $g$ for 20 min at 4°C and the supernatants were incubated with pre-equilibrated Glutathione-Sepharose 4B (GE Healthcare, Little Chalfont, UK) with rotation for 15 min at 4°C. The sepharose was then washed four times with wash buffer (50 mM $NaH_2PO_4$ pH 7.5, 1% NP-40, 500 mM NaCl, 10% glycerol, 1 mM DTT), two times with lysis buffer without DTT and equilibrated with storage buffer (20 mM Tris pH 7.5, 50% glycerol). A 25% sepharose slurry, with bound GST fusion proteins, was prepared with storage buffer and kept at −20°C.

Raji cells were centrifuged at 500 $g$ for 5 min, washed once with PBS, and resuspended at 5–10 × $10^6$ cells/ml in ice-cold protein extraction buffer (20 mM Tris pH 7.5, 0.5% NP–40, 100 mM NaCl,

2 mM $MgCl_2$) supplemented with 1x HALT Protease & Phosphatase Inhibitor Cocktail (Pierce, Thermo Fisher Scientific, Waltham, MA, USA). Cell lysis was facilitated by rotation for 15 min at 4°C after which cell extracts were cleared by centrifugation at 12,000 $g$ for 20 min at 4°C. Extract corresponding to 25–100 × $10^6$ Raji cells were used for each pulldown sample. Pre-equilibrated Glutathione-Sepharose 4B (GE Healthcare) with bound GST fusion proteins was incubated with protein extracts for 2–4 h with rotation at 4°C. The sepharose was then washed four times with 1 ml protein extraction buffer. Immobilized protein complexes were eluted by denaturation at 95°C for 10 min in 2x LDS Sample Buffer (Invitrogen, Thermo Fisher Scientific). For experiments with varied NaCl concentration, the sepharose was washed four times with protein extraction buffer containing 100, 150, 250, 500, or 1,000 mM NaCl followed by one additional wash step with 100 mM NaCl to equalize the beads before elution. For experiments including nucleases, 20 U/ml RNase ONE Ribonuclease (Promega) or 10 U/ml RQ1 RNase-Free DNase (Promega) were used in a modified protein extraction buffer with 5 mM $MgCl_2$, 5 mM $MnCl_2$, and 10 mM $CaCl_2$. Samples were stored at −20°C. Pulldown samples were run on a gel (see SDS–PAGE and Immunoblot), and suitable gel pieces were sent for MS analysis (Study 1).

**_In situ_ proximity ligation assays and confocal imaging**

Cells were washed briefly with PBS and fixed with 3.7% formaldehyde (Sigma-Aldrich) in PBS for 15 min. Samples were washed 3 × 10 min with PBS and incubated with block/permeabilization (B/P) buffer (PBS pH 7.3, 2% BSA, 0.02% Triton X-100; all from Sigma-Aldrich) for 30 min. Samples were then incubated for 90 min with combinations of primary antibodies: 1 µg/ml ARID1A (HPA005456; Sigma-Aldrich), 4 µg/ml BRG1 (H88) (sc-10768; Santa Cruz Biotechnology, Santa Cruz, CA, USA), 3 µg/ml DDIT3 (9C8) (ab11419; Abcam, Cambridge, United Kingdom), 2 µg/ml FUS (4H11) (sc-47711; Santa Cruz) diluted in B/P buffer. Control experiments were included by omitting one of two paired primary antibodies. Slides were washed 3 × 5 min in PBS with 0.1% Tween-20 with gentle agitation. _In situ_ proximity ligation assay (PLA) was performed with Duolink _In Situ_ Fluorescence products (OLink, Uppsala, Sweden) according to the manufacturer's instructions. Briefly, samples were incubated for 1 h at 37°C with a mixture of PLA probes anti-rabbit PLUS and anti-mouse MINUS, each diluted 1:5 in B/P buffer. Slides were washed 2 × 5 min in 1x Wash Buffer A (OLink), and subsequent ligation, wash steps, and amplification with Duolink _In Situ_ Detection Reagents Red (OLink) were prepared according to the manufacturer's instructions. Slides were mounted with cover slips using Duolink _In Situ_ Mounting Media with DAPI (OLink). A Zeiss LSM510 META confocal microscope system with LSM-5 software (Zeiss, Oberkochen, Germany) was used for confocal imaging. A 63x/1.4 oil objective and sequential scanning with excitation and META detector filter settings appropriate for each fluorophore was used (excitation 561 nm and BP600-710 for PLA signals, excitation 405 nm and BP420-475 for DAPI).

**Nuclear protein extraction and co-immunoprecipitation**

Cells from two confluent T75 or 15-cm petri dishes were harvested by scraping in PBS (Life Technologies) followed by centrifugation at

450 $g$ for 10 min at 4°C. The cell pellet (approximately 100–200 µl packed cell volume) was resuspended in 5 packed cell volumes of hypotonic lysis buffer (10 mM KCl, 10 mM Tris pH 7.5, 1.5 mM MgCl$_2$; all Life Technologies) supplemented with 1 mM DTT (Sigma-Aldrich) and 1x Halt Protease Inhibitor Cocktail (Thermo Scientific, Thermo Fisher Scientific) and was allowed to swell for 15 min on ice. The supernatant was discarded after centrifugation at 400 $g$ for 5 min at 4°C. Packed cells were resuspended in 2 packed cell volumes hypotonic lysis buffer and disrupted by 2-5 strokes of a syringe with a 27-gauge needle. The cytoplasmic fraction was removed after centrifugation at 10,000 $g$ for 20 min at 4°C. Pelleted nuclei were then resuspended in (2/3) packed cell volumes high-salt extraction buffer [0.42 M KCl, 10 mM Tris pH 7.5, 0.1 mM EDTA (all Life Technologies), 10% glycerol (Merck Chemicals, Merck)] supplemented with 1x Halt Protease Inhibitor Cocktail, and gently agitated in an icebox for 30 min. The nuclear fraction was collected after centrifugation at 20,000 $g$ for 5 min at 4°C and diluted to 150 mM salt concentration. A Benzonase (#71205, Merck Millipore, Merck) treatment with 5 U/ml for 15 min at 4°C was included during cell disruption for the Brg1 IP quantification experiments, the DDIT3 IP and the FLI1 IP.

Nuclear extracts were immunoprecipitated with Dynabeads Myone Streptavidin T1 (Thermo Fisher Scientific) using a protocol suitable for downstream Mass spectrometry analysis. The nuclear extract (60 µl, 100–200 µg) was diluted to 500 µl with IP wash buffer (150 mM KCL, 10 mM Tris pH 7.5, 0.1 mM EDTA, 10% glycerol) supplemented with 1x Halt Protease Inhibitor Cocktail and mixed with 10 µg antibody, either Brg1-biotin (ab200911, Abcam) or normal mouse IgG biotin (sc-2762, Santa Cruz Biotechnology). The nuclear extract/antibody mixture was incubated overnight at 4°C with gentle rotation. The next day, 75 µl beads (per reaction) were blocked in Rotiblock (Carl Roth, Karlsruhe, Germany) for approximately 30 min, followed by three washes with IP wash buffer. The nuclear extract/antibody mixture was then added to the beads and incubated for 2 h at 4°C with gentle rotation. The beads were washed 3 × 5 min on gentle rotation with 10 mM TEAB buffer (Thermo Fisher Scientific). Captured protein complexes were eluted twice with 20 µl 1% formic acid (Sigma-Aldrich) at 50°C, 500 rpm for 5 min followed by a third elution with 20 µl 2x LDS sample buffer with 10% sample reducing agent (Life Technologies) at 90°C, 500 rpm for 10 min to ensure that all bound proteins were released. After gel analysis, formic acid eluates were sent for MS analysis (Study 2). For the Brg1 IP quantification experiments, 100 µg nuclear extracts were used, the last three washes before elution were done in IP wash buffer, and samples were eluted in 2 × 50 µl 2x LDS sample buffer with 10% sample reducing agent. BRG1 IP replicates were scaled down two times. For the DDIT3 IP (DDIT3-biotin antibody, NB600-1335B, Novus Biologicals, Littleton, CO, USA) and the FLI1 IP (FLI1-biotin antibody, US biologicals 246159-biotin, Salem, MA, USA), the protocol was scaled up three times and samples were eluted in 2 × 75 µl or 2 × 150 µl 2x LDS sample buffer with 10% sample reducing agent. After gel analysis, LDS eluates from DDIT3 IP (Study 3) and FLI1 IP (Study 4) were sent for MS analysis. For quantification experiments, relative amounts of protein for each IP-sample were loaded on the gel, with consideration taken for dilutions during the immunoprecipitation procedure, in order to directly quantify the fraction of bound and non-bound protein.

## Mass spectrometry

Proteomic analyses were performed at The Proteomics Core Facility at the Sahlgrenska Academy, University of Gothenburg. Gel pieces (Study 1) were de-stained with 25 mM ammonium bicarbonate in 50% acetonitrile (ACN), in-gel digested by addition of 10 ng/µl trypsin (Pierce MS grade, Thermo Fisher Scientific) in 50 mM ammonium bicarbonate, and incubated overnight at 37°C. Peptides were extracted from the gel with 50% ACN in 1% acetic acid and dried down. Formic acid (Study 2) and LDS (Study 3 and 4) eluates from the immunoprecipitation (IP) were digested with trypsin using the filter-aided sample preparation (FASP) method [60]. Briefly, samples were reduced with 100 mM dithiothreitol at 60°C for 30 min, transferred to 30 kDa MWCO Pall Nanosep centrifugal filters (Sigma-Aldrich), washed with 8 M urea repeatedly, and alkylated with 10 mM methyl methane thiosulfonate. Digestion was performed in 50 mM TEAB, 1% sodium deoxycholate (SDC) buffer at 37°C by addition of 0.30 µg Pierce MS grade trypsin, and incubated overnight. An additional portion of trypsin was added and incubated for another 2 h. Peptides were collected by centrifugation, and SDC was removed by acidification with 10% trifluoroacetic acid. Samples were desalted using PepClean C18 spin columns (Thermo Fisher Scientific) according to the manufacturer's guidelines and dried down. The sample in Study 4 was processed through a HiPPR detergent removal spin column (Thermo Fisher Scientific) prior to C18 desalting. Samples were reconstituted in 3% ACN in 0.2% formic acid (FA).

Peptide samples were analyzed on a hybrid linear ion trap-FTICR mass spectrometer equipped with a 7T ICR magnet (LTQ-FT, Study 1), an Orbitrap Fusion Tribrid mass spectrometer (Study 2 and 3), or a Q Exactive HF (Study 4) mass spectrometer (all Thermo Fisher Scientific) interfaced with an Easy nLC 1000 liquid chromatography system. Peptides were trapped and separated using an C18 precolumn (45 × 0.075 mm I.D.) and analytical column (300 × 0.075 mm I.D) packed with 3 µm Reprosil-Pur C18-AQ particles using a gradient from 5 to 80% ACN in 0.2% FA for 40 min. MS spectra and MS/MS spectra were acquired in the FTICR and the LTQ-trap, respectively. For each scan of FTICR, the three most intense double or triple charged ions were sequentially fragmented in the linear trap by collision-induced dissociation (CID). In study 2 and 3, precursor ion mass spectra were acquired at 120,000 resolution and MS/MS analysis was performed in a data-dependent mode where CID spectra of the most intense precursor ions were recorded in ion trap at 30,000 resolution and collision energy setting of 30 for 3 s ("top speed" setting). Charge states 2–7 were selected for fragmentation, dynamic exclusion was set to 45 s. In Study 4, the precursor ion mass spectra were acquired at 60,000 resolution and MS/MS analysis was performed in a data-dependent mode where HCD spectra of the Top 10 most intense precursor ions were recorded at 30,000 resolution. Charge states 2–4 were selected for fragmentation with collision energy setting of 28 and dynamic exclusion was set to 20 s.

Data analysis was performed utilizing Proteome Discoverer version 1.4 (Thermo Fisher Scientific) against the Human Swissprot Database version 55.3, 2009 (Study 1, verified against version Nov 2017 for data upload), May 2016 (Study 2), Sep 2016 (Study 3), and March 2017 (study 4). Mascot (2.3 or 2.5.1, Matrix Science) was used as a search engine with precursor mass tolerance of 5 ppm and fragment mass tolerance of 500 mmu. Tryptic peptides were

accepted with one missed cleavage, variable methionine oxidation and static cysteine propionamide modifications (Study 1) or zero to one missed cleavages and variable methionine oxidation, static cysteine methylthio modifications (Study 2, 3, and 4). The detected peptide threshold in the software was set to a significance level of Mascot 95% (Study 1) or Mascot 99% (Study 2, 3, and 4) by searching against a reversed database, and identified proteins were grouped by sharing the same sequences to minimize redundancy.

## Whole-cell extraction for histone modification analysis

Whole-cell extracts were prepared on ice with cells from 70 to 95% confluent 10- or 15-cm petri dishes (stable clones) or 6-well plates (after transient transfection) by scraping in PBS followed by centrifugation at 450 $g$ for 10 min at 4°C. The cell pellet was lysed in RIPA buffer (25 mM Tris•HCl pH 7.6, 150 mM NaCl, 1% NP-40, 1% sodium deoxycholate, 0.1% SDS, Pierce, Thermo Scientific) with 5 mM EDTA and 1x Halt Protease Inhibitor Cocktail and incubated on ice for 10 min with gentle mixing every 5 min. The lysate was then sonicated in order to disrupt viscous DNA. In order not to lose any histone proteins remaining in the insoluble fraction, no centrifugation was done at this point. Samples were then mixed with NuPAGE 4x LDS Sample Buffer to final concentrations of 106 mM Tris–HCl, 141 mM Tris Base, 2% LDS, 10% glycerol, 0.51 mM EDTA, 0.22 mM SERVA Blue G250, and 0.175 mM Phenol Red, pH 8.5, and heated at 95°C for 10 min before loading on gels. Equal protein amounts were analyzed with immunoblot to evaluate the amount of histone 3 lysine 4 trimethylation (H3K4me3), histone 3 lysine 27 trimethylation (H3K27me3), and histone 3 lysine 27 acetylation (H3K27Ac) using histone H4 as a loading control, see immunoblot.

## SDS–PAGE and immunoblot

Protein samples were size-separated with SDS–PAGE using the Novex NuPAGE system (Life Technologies) according to the manufacturer's instructions. In short, protein extracts were mixed with 1x NuPAGE LDS sample buffer and 10% NuPAGE sample reducing agent, denatured at 70°C for 10 min, and separated on NuPAGE 4–12% Bis-Tris or 3–8% Tris-acetate gels. Separated proteins were stained with SimplyBlue SafeStain (Life Technologies) or transferred to polyvinylidene difluoride membranes (0.45 μm, Life Technologies) by wet blot. The membranes were blocked with 5% skim milk (Merck Chemicals) or 5% BSA (Sigma-Aldrich) in TBS-T buffer (50 mM Tris-HCl pH 6.8, 50 mM NaCl, 0.1% Tween-20; all from Sigma-Aldrich). Membranes were incubated overnight at 4°C with 0.5 μg/ml ARID1A (PSG3) (sc-32761; Santa Cruz), 0.5 μg/ml ARID1A (HPA005456; Sigma-Aldrich), 1.1 μg/ml BAF 47 (Ab12167; Abcam), 0.5 μg/ml BAF60A (sc-135843; Santa Cruz), 0.5 μg/ml BAF 155 (DXD7) (sc-32763; Santa Cruz), 0.5 μg/ml BAF 170 (E-6) (sc-17838; Santa Cruz), 0.5 μg/ml BRG1 (H-88) (sc-10768; Santa Cruz), 0.2 μg/ml BRG1 (G-7) (sc-17796; Santa Cruz), 0.7 μg/ml DDIT3 (15204-1; Proteintech, Chicago, IL, USA), 0.2 μg/ml EWS (sc-28327; Santa Cruz), 0.5 μg/ml EZH2 (#07-689; Merck Millipore), 0.5 μg/ml FLI1 (Ab15289; Abcam), 0.2 μg/ml FUS (4H11) (sc-47711; Santa Cruz), 1 μg/ml GAPDH (ab9484; Abcam), 0.5 μg/ml GFP (JL-8) (632381; Clontech), 1 μg/ml H3K4me3 (#05-745R; Merck Millipore), 0.1 μg/ml H3K27ac (ab177178; Abcam), 0.5 μg/ml H3K27me3 (#07-449; Merck Millipore), 1:10,000–30,000 (concentration unavailable)

histone H4 (#04-858; Merck Millipore) or 1 μg/ml SS18 (H80) (sc-28698; Santa Cruz), followed by 1-h incubation with anti-mouse or anti-rabbit HRP-conjugated secondary antibody (32430 and 32460; Thermo Scientific) at room temperature. Protein detection via luminescent signals was captured with ImageQuant LAS 4000 mini (GE Healthcare Life Sciences) after incubation with SuperSignal West Dura Extended Duration Substrate or SuperSignal West Femto Max Sensitivity Substrate (Thermo Scientific). Some membranes were stripped with ReBlot Plus (2504, Merck Millipore) during 15-min incubation in room temperature before relabeling the membrane with another primary antibody. Bands were quantified using Multi-Gauge V3.2 (Fujifilm, Tokyo, Japan).

## Statistical analysis

Quantification values of immunoblots for the 3–4 replicates are presented as means ± SEM with each individual experiment indicated. GraphPad Prism software (version 7.00, GraphPad, San Diego, CA, USA) was used for statistical analysis. The Student's $t$-test (unpaired, two-sided $t$-test) was used with $\alpha < 0.05$ considered significant. Original data for all quantifications are shown in the source data.

## RNA-sequencing

The Smart-seq2 protocol [61] was used to generate sequencing libraries from HT1080 wt ($n = 4$), HT1080 EGFP ($n = 4$), HT1080 FUS-DDIT3-EGFP ($n = 3$), and HT1080 EWSR1-FLI1-EGFP ($n = 4$). Adherent cells were washed with DPBS and scraped directly in RLT lysis buffer (Qiagen, Hilden, Germany) supplemented with β-mercaptoethanol (MP Biomedicals). Total RNA was extracted using the RNeasy Micro Kit with DNase treatment (Qiagen) according to the manufacturer's recommendations and stored at −80°C. The RNA quality was confirmed using Agilent RNA 6000 Nano Kit on a 2100 BioAnalyzer Instrument (Agilent Technologies, Santa Clara, CA, USA).

Reverse transcription was performed on 10 ng total RNA. An initial hybridization step was performed by adding 1 mM dNTP and 1 μM biotinylated adapter sequence-containing oligo-dT30VN (5′-biotin-AAGCAGTGGTATCAACGCAGAGTACT30VN-3′) to the sample (both Sigma-Aldrich, concentrations refer to the final reverse transcription reaction) and incubating at 72°C for 3 min. Subsequently, 1x first-strand buffer (50 mM Tris–HCl pH 8.3, 75 mM KCl, and 3 mM MgCl$_2$), 5 mM dithiothreitol (both Invitrogen, Thermo Fisher Scientific), 10 mM MgCl$_2$ (Ambion, Thermo Fisher Scientific), 1 M betaine (Sigma-Aldrich), 0.6 μM biotinylated adapter sequence-containing template switching oligonucleotide (5′-biotin-AAGCAGTGGTATCAACGCAGAGTACATrGrG+G-3′ with rG = riboguanosine and +G = locked nucleic acid modified guanosine, Eurogentec, Liège, Belgien), 15 U RNaseOUT, and 150 U SuperScript II (both Invitrogen, Thermo Fisher Scientific) were added resulting in 15 μl reaction volume. Reverse transcription was run at 42°C for 90 min and 70°C for 15 min in a T100 instrument (Bio-Rad, Hercules, CA, USA). cDNA was stored at −20°C.

Preamplification was performed by mixing 7.5 μl cDNA sample with 1x KAPA Hifi HotStart Ready Mix (KAPA Biosystems, Wilmington, MA, USA), 0.1 μM primer (5′-AAGCAGTGGTATCAACGC AGAGT-3′; Sigma-Aldrich) in a reaction volume of 50 μl. Preamplification was run at 98°C for 3 min followed by 24 cycles of

amplification at 98°C for 20 s, 67°C for 15 s, and 72°C for 6 min, and a final additional incubation at 72°C for 5 min in a T100 instrument. Samples were transferred from 72°C directly to dry ice and stored at −20°C. Purification of samples was performed using Agencourt AMPure XP beads (BD Biosciences).

The 50 μl sample was mixed with 40 μl beads (beads-to-sample ratio of 0.8) followed by incubation at room temperature for 5 min on the bench and 5 min on a magnet (DynaMag, Thermo Fisher Scientific). Supernatant was discarded, and beads were washed twice with 200 μl 80% ethanol and left to dry. Elution of samples was performed with 17.5 μl RNase/DNase-free water (Invitrogen, Thermo Fisher Scientific) by incubation at room temperature for 2 min on the bench and 2 min on the magnet. Quality and concentration measurement was performed with Agilent High Sensitivity DNA Kit on a 2100 Bioanalyzer Instrument (Agilent Technologies), and 100 pg of cDNA was used for tagmentation and indexing.

Tagmentation and indexing were performed using Nextera XT DNA Library Preparation Kit and Nextera XT Index Kit v2 (Illumina, San Diego, CA, USA). First, 10 μl Tagment DNA Buffer and 5 μl Amplicon Tagment Mix was added to 5 μl sample, and tagmentation was run at 55°C for 5 min in a T100 instrument. Next, 5 μl neutralize tagment buffer was added followed by centrifugation for 1 min at 1100 rpm (LMC-3000, rotor R-2, Biosan, Riga, Latvia) and 5-min incubation at room temperature. For indexing and library amplification, 15 μl Nextera PCR Master Mix and 5 μl of each index 1 (i7) and index 2 (i5) adapters were added and amplification was run at 72°C for 3 min, 95°C for 30 s followed by 16 cycles of amplification at 95°C for 10 s, 55°C for 30 s, and 72°C for 30 s, and a final additional incubation at 72°C for 5 min in a T100 instrument. Purification of samples was performed using Agencourt AMPure XP beads as before but using a beads-to-sample ratio of 0.6 by adding all sample volume to 30 μl beads.

The concentration of each sample was analyzed using Qubit dsDNA High Sensitivity Assay Kit (Invitrogen, Thermo Fisher Scientific). Library quality control and size was ensured by capillary gel-electrophoresis on a Fragment Analyzer using the DNF-474 High Sensitivity NGS kit (both Agilent technologies). Libraries were pooled equimolarly based on Fragment Analyzer data, and the final pool was quantified by qPCR using the NEBNext Library Quantification kit (New England BioLabs, Ipswich, MA, USA). The libraries were clustered at 1.8 pM supplemented with 1% PhiX control on a MiniSeq instrument (both Illumina) using paired-end sequencing with a read-length of 2 × 75 bp.

Alignment of Illumina reads was performed using STAR RNA-seq aligner v2.6 [62] using ENSEMBL GRCh38 assembly as the reference genome. Read count matrices were generated using the HTSeq python framework v0.9.1. [63]. Genes with a total count number < 10 were excluded from downstream analyses. Differential expression was analyzed using the R package DESeq2, based on shrink estimation for dispersion and fold-change using a negative binomial distribution model [64]. Adjusted *P*-values were calculated using the Benjamini–Hochberg method. Genes at least twofold regulated (adjusted *P*-value ≤ 0.05) were analyzed in downstream analysis using the molecular signature database (MSigDB) v6.2 [65,66]. Gene lists were compared to the gene-set collection "chemical and genetic perturbations" (GSEA: http://software.broadinstitute.org/gsea/msigdb/index.jsp), and top 20 gene-sets (FDR *q*-value < 0.05) for each comparison were selected.

## Data availability

The mass spectrometry proteomics data from this publication have been deposited to the ProteomeXchange Consortium via the PRoteomics IDEntifications database (PRIDE) [67] [https://www.ebi.ac.uk/pride/archive/] and assigned the dataset identifier PXD012680. The RNA-sequencing data from this publication have been deposited in NCBI's Gene Expression Omnibus (GEO) database [68] [https://www.ncbi.nlm.nih.gov/geo/] and assigned the identifier GSE125941.

**Expanded View** for this article is available online.

## Acknowledgements

This study was supported by the Swedish Cancer Society (CAN2015/7130, 2016/438), The Swedish state under the agreement between the Swedish government and the county councils, the ALF agreement (ALFGBG-722211 and 716321), Knut and Alice Wallenberg Foundation, Wallenberg Centre for molecular and translational medicine at University of Gothenburg, the Swedish Research Council (2017-01392), VINNOVA, the Swedish Society for Medical Research, the Assar Gabrielssons Foundation, the BioCARE National Strategic Research, the Swedish Childhood Cancer Foundation (PR2017-0043), the Johan Jansson Foundation for Cancer Research, the Swedish Society of Medicine, and the Wilhelm and Martina Lundgren Foundation for Scientific Research. We thank the Centre for Cellular Imaging at the Sahlgrenska Academy, University of Gothenburg, for help with imaging and Jacqueline Forzelius at #explainartist for graphical presentation. The Proteomics Core Facility at Sahlgrenska Academy, Gothenburg University, performed the mass spectrometry analysis for protein identification. We are grateful of Inga-Britt and Arne Lundbergs Forskningsstiftelse for the donation of the Orbitrap Fusion Tribrid MS instrument.

## Author contributions

ML and CT contributed with the main laboratory work, planned and designed experiments, prepared figures and other result presentations, and took a main part in writing the manuscript. PG made the histone analysis and associated cell culture/cloning work and wrote corresponding parts of the manuscript. EJ helped with cell culture work, data handling, and formatting and helped writing the manuscript. DA and SD contributed with manuscript writing and figure preparation; RR made important lab work and helped writing the manuscript. CV added the EZH2 analysis and helped writing the manuscript. MLS helped with bioinformatics analysis of RNA-seq data. HF and AS contributed with critical data interpretation and writing of the manuscript. PÅ contributed with project planning, experimental design, data handling and interpretation, and writing of the manuscript.

### Conflict of interest

The authors declare that they have no conflict of interest.

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
