## [Review Process File · EMBO Reports]

FET family fusion oncoproteins target the SWI/SNF chromatin remodeling complex

Malin Lindén, Christer Thomsen, Pernilla Grundevik, Emma Jonasson, Daniel Andersson, Rikard Runnberg, Soheila Dolatabadi, Christoffer Vannas, Manuel Luna Santamaría, Henrik Fagman, Anders Ståhlberg and Pierre Åman

Review timeline:

Submission date:	14 January 2018
Editorial Decision:	7 February 2018
Revision received:	7 June 2018
Editorial Decision:	28 June 2018
Revision received:	31 July 2018
Editorial Decision:	22 August 2018
Revision received:	15 February 2019
Editorial Decision:	4 March 2019
Revision received:	6 March 2019
Accepted:	11 March 2019

Editor: Achim Breiling

Transaction Report:

1st Editorial Decision

7 February 2018

Thank you for the submission of your research manuscript to EMBO reports. We have now received reports from the referees that were asked to evaluate your study, which can be found at the end of this email.

As you will see, all referees think the manuscript is of interest, but requires major revision to allow publication in EMBO reports. All referees have a number of concerns and/or suggestions to improve the manuscript, which we ask you to address in a revised manuscript. As the reports are below, I will not detail them here. However, we think that it will be of great importance to provide further functional data (as pointed out by all referees, in particular by referee #2 in major point 2), which should strengthen the physiological relevance of the findings, and that in particular the effects of the fusion proteins on histone modifications need further and stronger experimentation (referee #1 and major point 2 by referee #3). Further, please make sure that appropriate controls and statistics are used throughout the manuscript (see the report of referee #1), and also consider the technical note of referee #2 on histone extraction (at the end of her/his report).

Given the constructive referee comments, we would like to invite you to revise your manuscript with the understanding that all referee concerns must be addressed in the revised manuscript and in a point-by-point response. Acceptance of your manuscript will depend on a positive outcome of a second round of review. It is EMBO reports policy to allow a single round of revision only and acceptance or rejection of the manuscript will therefore depend on the completeness of your responses included in the next, final version of the manuscript.

Revised manuscripts should be submitted within three months of a request for revision; they will otherwise be treated as new submissions. Please contact us if a 3-months time frame is not sufficient for the revisions so that we can discuss the revisions further.

Supplementary/additional data: The Expanded View format, which will be displayed in the main HTML of the paper in a collapsible format, has replaced the Supplementary information. You can submit up to 5 images as Expanded View. Please follow the nomenclature Figure EV1, Figure EV2 etc. The figure legend for these should be included in the main manuscript document file in a section called Expanded View Figure Legends after the main Figure Legends section. Additional Supplementary material should be supplied as a single pdf labeled Appendix. The Appendix includes a table of content on the first page, all figures and their legends. Please follow the nomenclature Appendix Figure Sx throughout the text and also label the figures according to this nomenclature.

For more details please refer to our guide to authors:
<http://embor.embopress.org/authorguide#manuscriptpreparation>

See also our guide for figure preparation:
http://www.embopress.org/sites/default/files/EMBOPress_Figure_Guidelines_061115.pdf

Important: All materials and methods should be included in the main manuscript file.

Please add a conflict of interest statement, author contributions to the manuscript text (next to the acknowledgements), and a short running title and up to 5 keywords to the title page.

Regarding data quantification and statistics, can you please specify, where applicable, the number "n" for how many independent experiments (biological replicates) were performed, the bars and error bars (e.g. SEM, SD) and the test used to calculate p-values in the respective figure legends. Please provide statistical testing where applicable. See:
<http://embor.embopress.org/authorguide#statisticalanalysis>

We now strongly encourage the publication of original source data with the aim of making primary data more accessible and transparent to the reader. The source data will be published in a separate source data file online along with the accepted manuscript and will be linked to the relevant figure. If you would like to use this opportunity, please submit the source data (for example scans of entire gels or blots, data points of graphs in an excel sheet, additional images, etc.) of your key experiments together with the revised manuscript. Please include size markers for scans of entire gels, label the scans with figure and panel number, and send one PDF file per figure.

- a complete author checklist, which you can download from our author guidelines (<http://embor.embopress.org/authorguide#revision>). Please insert page numbers in the checklist to indicate where the requested information can be found.
- a letter detailing your responses to the referee comments in Word format (.doc)
- a Microsoft Word file (.doc) of the revised manuscript text
- editable TIFF or EPS-formatted single figure files in high resolution (for main figures and EV figures)
- single files of EV tables or datasets

I look forward to seeing a revised version of your manuscript when it is ready. Please let me know if you have questions or comments regarding the revision.

REFeree REPORTS

Referee #1:

Thomsen et al. report interaction between members of the FUS/EWS/TAF15 (FET) family of proteins and of their oncogenic fusion variants FUS-DDIT3, EWS-FLI1 and EWS-ERG with the BAF nucleosome remodeling complex. Based on pulldown experiments with the N-terminal FET domain as a bait followed by mass spec analysis they identified interaction with almost the full set of BAF complex components for all three FET family members. They validated their results using immunoblotting and proximity ligation assays, and delineated the minimal domain required for BAF interaction to a conserved 26 amino acid FET binding motif. They further demonstrate that FET fusion proteins in myxoid liposarcoma and Ewing

sarcoma are more readily and firmly bound to the BAF complex than wildtype FET proteins. These results are convincing and confirm recently published observations in Ewing sarcoma suggesting recruitment and tethering of the BAF complex to de novo enhancers by EWS-FLI1, and expand them to other FET fusion proteins.

However, the knowledge gain from this study remains moderate, as the functional consequences of FET-fusion protein interaction with the BAF complex remain still elusive. The authors demonstrate that the fraction of total wildtype FET proteins tethered to the BAF complex is small in myxoid liposarcoma and Ewing sarcoma cell lines, while most of their fusion proteins were found associated with it. Do FET fusion proteins compete for binding of FET wildtype proteins to the complex? This may be answered by performing knockdowns of the fusion proteins in the tumor cell lines and testing the BAF complex bound fraction of wildtype FUS and EWS. Moreover, the authors overexpressed EGFP-fusions of FUS-DDIT3 and EWS-FLI1 in a FLI1 expressing fibrosarcoma cell line HT1080 and observed slightly increased levels of the repressive histone mark H3K27me3. While this finding is attractive as it may potentially unravel a conserved mechanism of oncogenesis through impairment of the antagonistic role of the BAF complex to the PRC2 gene silencing complex, the experiment lacks the appropriate control and statistics. Parental HT1080 cells instead of EGFP expressing cells were used as negative controls, histone H4 but not total H3 was used as loading control. It is unclear, how big the fraction of transfected cells was, if EGFP was used to sort out transfected cells, and if EGFP toxic effects such as ROS production (for review see Ansari et al., PMID:27435468) might have been the reason for the observed moderate signal shift. As previous reports did not observe changes in the repressive H3K27me3 mark upon knockdown of EWS-FLI1 in Ewing sarcoma cell lines, but instead a strong decrease in the activating H3K27ac mark, clarification of the functional consequences of FET fusion oncogene interaction with BAF on enhancer activity requires a thorough investigation of this issue. Consequently, the authors should also test for H3K27ac levels in their HT1080 model system.

Referee #2:

The study by Thomsen et al describes the physical interaction between the FET family of transcription factors and members of the SWI/SNF remodeling complex.

The FET family of proteins consists of FUS, EWSR1 and TAF15 and are normally expressed in all tissues and engage at different stages of gene expression. The human FET proteins are associated with transcription, splicing, microRNA processing, RNA transport, signaling and maintenance of genomic integrity. Fusion proteins containing the N-terminal domain of FETs and the C-terminal domain of DNA binding proteins are commonly found in many sarcomas and have been shown to be the driving force for carcinogenesis in these types of cancer.

The mammalian SWI/SNF complex is an ATP-dependent chromatin remodeler composed of 12-15 core subunits that regulate genomic architecture and DNA accessibility. Recent studies revealed that genes encoding the different SWI/SNF subunits are collectively mutated in over 20% of human cancers and that specific subunits are mutated in different cancer types suggesting tissue specific functions.

Using GST-tagged recombinant constructs as baits in pulldown experiments with sarcoma cell line extracts and mass spectrometry, the authors demonstrate that the N-terminal domain of all 3 FETs can strongly interact with members of the mammalian SWI/SNF complex. Moreover, the authors use a proximity ligation assay approach to confirm that both the wild type FUS and the fusion FUS-DDIT3 proteins interact with SWI/SNF components ARID1A and BRG1 *in vivo*, in the same sarcoma cell lines used for Mass Spec. By generating several truncated GST-tagged constructs of the N-terminal domain of FETs, they specifically identify the area of the FETs responsible for the interaction with SWI/SNF. Furthermore, the authors perform a number of co-IP experiments for BRG1 ("reverse" IP) to confirm the interaction with wild type or fusion FETs and differential salt extraction to demonstrate the strength of the interaction for each SWI/SNF component identified. Finally, in order to get mechanistic insight into the effect that these FET-SWI/SNF interactions have on SWI/SNF function, the authors perform immunoblot analysis for the H3K27me3 modification in cell lines expressing either wild type or fusion FETs and find that the levels of this modifications are elevated in the fusion expressing cells. Given that SWI/SNF antagonizes the polycomb repressive complex 2 (PRC2) which catalyzes the H3K27me3 mark, the authors suggest that the H3K27me3 levels correlate with perturbed SWI/SNF function.

From the present analysis, the authors propose 2 major findings, the first that the N-terminal domain of all 3 FET proteins interacts with many members of the SWI/SNF complex and the second, that this interaction disrupts the normal role of SWI/SNF to oppose the PRC2 suppressive complex and thus this interaction leads to increased EZH2 enzymatic activity and increased H3K27me3 levels.

Given the importance of the FET fusion onco-proteins as driving events in sarcomas, the poor prognosis and difficulty of treatment of these cancers and the importance of the SWI/SNF complex for transcription, the identification of the FET-SWI/SNF interaction is of high importance.

The manuscript is well written and, at least for Figures 1-3, the data is fairly robust and supportive of the conclusions presented. However, the data presented in Figure 4 and Supplementary Figure 3 are weak and correlative. In this reviewer's opinion, while the experiments are generally well performed, the novelty is minimal and the functional insight very weak and correlative (Figure 4), reducing enthusiasm for publishing the paper in EMBO reports.

More specifically, the major concerns are:

1. Lack of novelty.

A recently published study (Boulay et al. Cell 171, 2017) demonstrated in detail how the Ewing sarcoma fusion protein EWS-FLI1 interacts with many members of the SWI/SNF complex and provided functional insight for the altered targeting of SWI/SNF in this setting.

In the present study, the authors conduct a more thorough analysis in terms of the specific domains of FET proteins that interact with SWI/SNF members and demonstrate the interaction in different types of sarcomas other than Ewing but nevertheless, there is significant overlap with the Boulay et al 2017 paper.

2. Weak functional analysis

Although one of the major roles of SWI/SNF is to oppose the function of PRC2, the levels of H3K27me3, especially with the limitations of detection of immunoblot, are not the best readout of SWI/SNF function. A number of recent reports have demonstrated that SWI/SNF binds to enhancer elements and regulate the H3K27ac mark and the accessibility of chromatin for transcription in these areas. In this sense, alteration in global levels of H3K27ac is a better readout of SWI/SNF perturbed function. Furthermore, even if H3K27me3 was a reliable readout of SWI/SNF function, the change in global levels as seen in Figure 4 between HT1080 expressing either wild type FUS or fusion FUS-DDIT3, is marginal and in no way, supports a perturbed SWI/SNF function.

On a technical note, the cell lysis and protein extraction method used for histone modification analysis is not the appropriate one as it is known that the majority of histone proteins precipitate in an insoluble fraction. A more specific histone extraction protocol is needed in order to extract the majority of histone proteins from chromatin and have confidence that one has a representative cell extract.

Referee #3:

In this work Thomsen, Lindén et al. describe the direct physical interaction between the N-terminal part of the FET protein members EWSR1, FUS and TAF15 and the SWI/SNF chromatin-remodeling complex. Of particular interest, they show how the FUS-DDIT3, EWSR1-FLI1 and EWSR1-ERG translocations display a stronger interaction with the SWI/SNF complex, compared to their wild type counterparts, which lead to a functional change in SWI/SNF activity, increasing the deposition of the H3K27me3 histone modification. The authors suggest that the difference in the interaction strength between the wild-type and translocated FET proteins may underlie, at least in part, the oncogenic role of FUS and EWSR1 N-terminal moieties in Mioxoid Liposarcoma and Ewing sarcoma.

The manuscript is well written and the experimental approach, the rationale and the results are clearly presented, and appropriately supported by the experimental data. The topic is very timely and addresses a major question in field on sarcoma pathogenesis, namely, the role of the N-terminal portion of FET proteins in the translocations involved in a wide range of tumors, and the potentially critical functional differences with their wild-type counterparts. Mutations in SWI/SNF protein family members have been associated with more than 20% of human malignancies, and may represent a key oncogenic event in human cancer, offering a new promising Achilles' heel for some of these entities. In translocated human sarcomas, particularly Ewing sarcoma (expressing the EWSR1-FLI1 fusion protein), the role of the EWSR1 N-

terminal fragment has been recently suggested to play a critical role in allowing prion-like aggregation of EWSR1-FLI1 with wild type EWSR1 complexes, which are indispensable to activate the oncogenic program sustaining tumor growth. The present work confirms and expands this knowledge to other tumor entities, including Mixoid Liposarcoma (expressing the FUS-DDIT3 translocation), and suggests that a similar mechanism may be active in a broader panel of human sarcomas, driven by chromosomal translocations involving FET proteins.

Although the biochemical characterization of the interaction between FET proteins and the SWI/SNF complex is very detailed and convincing, the reviewer would appreciate some additional evidences of the role of this interaction in tumor development to be added to the current work.

Major points:

1) In Figure 2 the authors show the importance of the 31-66 amino acidic portion of the FUS protein in the interaction with SWI/SNF complex. Along with the difference in binding strength between the wild type and translocated FET proteins, this is a major point of the work. It would be important to add some functional evidences to the role of these mutants in the development of Mixoid Liposarcoma. First of all, the authors should create at least some of these mutants for the FUS-DDIT3 fusion protein, and add the mutant carrying the 31 to 66 amino acidic part of FUS, fused to DDIT3. These mutants could be very useful in determining the importance of the SWI/SNF-FUS interaction for the transforming ability of FUS-DDIT3. Second, by expressing these mutants in unrelated cell types (like SW872 or HT1080 cell lines) they could assess if the mutants are still able to bind to the DNA and activate some of the reported target genes (MMP2, MMP9, Pentaxin 3 and Fibronectin 1). Moreover, since the FUS-DDIT3 translocation has been reported to be able to transform NIH 3T3 cells in vitro and in vivo, the authors could express the wild type FUS-DDIT3 fusion protein along with the mutants and assess the changes in transforming ability (soft agar or tumorigenicity assays in immunosuppressed mice).

2) In Figure 4, the authors claim that the expression of the FUS-DDIT3 and EWSR1-FLI1 proteins in HT1080 cells result in an increase in the deposition of the H3K27me3 repressive histone mark. These data are in conflict with two recent reports describing changes in H3K27ac mark but not H3K27me3 (as noted by the authors). Since the induction of H3K27me3 deposition is relatively subtle, and the timing of the H3K27me3 changes extremely rapid (24 to 48 hours after transfection, which is not a classical time frame for these type of chromatin remodeling events), the reviewer feels that a confirmation of these results using a different approach could be important. One possibility would be a detailed Mass Spectrometry analysis of histone H3 modifications, with a quantitative determination of the levels of H3K27me3 in wild type versus FUS-DDIT3 and EWSR1-FLI1-expressing cells. In addition, since the EWSR1-FLI1 protein has been shown to induce the transcription of EZH2, one of the core members of the PRC2 complex involved in H3K27me3 deposition, the authors should consider the possibility of an indirect effect, not mediated by a competition with the SWI/SNF complex, but by an increase of the PRC2 activity. To this end, the authors should verify if in their cellular model if the expression of FUS-DDIT3 and EWSR1-FLI1 results in the increase of the PRC2 protein members EZH2, EED and SUZ12. This should be performed on a longer time point using a stable lentiviral approach.

Minor points:

1) In Figure 3A the size of the EWSR1-FLI1 protein seems to be different between the first and last lanes.

1st Revision - authors' response

7 June 2018

Referee #1:

Thomsen et al. report interaction between members of the FUS/EWS/TAF15 (FET) family of proteins and of their oncogenic fusion variants FUS-DDIT3, EWS-FLI1 and EWS-ERG with the BAF nucleosome remodeling complex. Based on pulldown experiments with the N-terminal FET domain as a bait followed by mass spec analysis they identified interaction with almost the full set of BAF complex components for all three FET family members. They validated their results using immunoblotting and proximity ligation assays, and delineated the minimal domain required for BAF interaction to a conserved 26 amino acid FET binding motif. They further demonstrate that FET fusion proteins in myxoid liposarcoma and Ewing

sarcoma are more readily and firmly bound to the BAF complex than wildtype FET proteins. These results are convincing and confirm recently published observations in Ewing sarcoma suggesting recruitment and tethering of the BAF complex to de novo enhancers by EWS-FLI1, and expand them to other FET fusion proteins.

#1 However, the knowledge gain from this study remains moderate, as the functional consequences of FET-fusion protein interaction with the BAF complex remain still elusive.

The news value of this manuscript: Our data is the first to point out all the FET NTDs as mediators of SWI/SNF interactions in this large family of oncoproteins. The present manuscript shows that the whole family of FET-proteins and not only one member, as EWS-FLI1 in the Boulay paper, has capacity to bind SWI/SNF. We define the part of the FET-NTD that is required for this binding and also show different binding profiles for normal and oncogenic FET proteins (FUS-DDIT3 and EWS-FLI1). Of interest for understanding the oncogenic activity of FET oncoproteins is also the highly divergent DNA binding properties of FUS-DDIT3 and EWSR1-FLI1, suggesting that other mechanisms than recruitment of SWI/SNF to novel binding sites are of central importance.

Functional effects of the FET-SWI/SNF interactions are up to this time point hard to study. Except for genomic studies employing ChIPSEQ and similar techniques, we are not aware of any functional tests for the human SWI/SNF complex. Genomic effects were not the focus of this investigation.

#2 The authors demonstrate that the fraction of total wildtype FET proteins tethered to the BAF complex is small in myxoid liposarcoma and Ewing sarcoma cell lines, while most of their fusion proteins were found associated with it. Do FET fusion proteins compete for binding of FET wildtype proteins to the complex? This may be answered by performing knockdowns of the fusion proteins in the tumor cell lines and testing the BAF complex bound fraction of wildtype FUS and EWS.

Thanks for this very interesting comment and suggestion! The wild type FET proteins and the SWI/SNF complex are very abundant in the cells while the oncogenic proteins are expressed at much lower levels mainly due to much shorter protein half-life (Åman et al 2016 added reference). The low expression of FET oncoproteins and severe difficulties in reliably transfecting/knocking FUS-DDIT3 down in MLS cell lines without off target effects, made us instead choose to over-express FUS-DDIT3 in our experimental system and quantify the amounts of normal and oncogenic proteins co-precipitated with anti BRG1-SWISNF. These experiments gave unexpected, but highly interesting results that are now included in the Figure 3, discussed in the results/discussion part, and lead us to partially revise/develop our model for how FET proteins and SWI/SNF may interact.

#3 Moreover, the authors overexpressed EGFP-fusions of FUS-DDIT3 and EWS-FLI1 in a FLI1 expressing fibrosarcoma cell line HT1080 and observed slightly increased levels of the repressive histone mark H3K27me3. While this finding is attractive as it may potentially unravel a conserved mechanism of oncogenesis through impairment of the antagonistic role of the BAF complex to the PRC2 gene silencing complex, the experiment lacks the appropriate control and statistics. Parental H1080 cells instead of EGFP expressing cells were used as negative controls.

As suggested by the referee, we have included new experiments for evaluation of histone modifications. We included HT1080 GFP as a control and analyzed the histone modifications in cell lines with stable FET-oncogene expression in 2 biological replicates and not only in transiently transfected cells. Most notably: each of the experiments both with stable and transient transfections (Figure 4A and Figure EV2), show an increased H3K27 trimethylation in FET oncogene expressing cells.

The increased methylation changes could be an effect of increased levels of PRC2 complex as the amount of EZH2 correlate with the H3K27 methylation levels in the stable transfection experiments. Increased EZH2 levels are also compatible with an impaired SWI/SNF function as reported by other groups (See references in manuscript).

Also, the “slight increase in H3K27me3 levels” may seem small but considering the size of the genome it may well translate to a changed regulation of hundreds or, with downstream effects, even thousands of genes!

#4 histone H4 but not total H3 was used as loading control. It is unclear, how big the fraction of transfected cells was, if EGFP was used to sort out transfected cells.

The fraction of GFP or DsRED positive cells in transient transfections in HT1080 cells is routinely above 80% and the number of positive cells were monitored in the transient transfection experiments. The fusion oncogenes are toxic and any analysis has to be made within the first 48 hours before high expressing cells show signs of stress and die or go senescent. For stable transfectants, GFP positive cells were sorted out by FACS and seeded out at 0,5 cells per well in 96-well plates with feeder fibroblasts. Single cell colonies

were picked, expanded and tested for oncoprotein expression by western blot. Only cells/clones with very low oncoprotein expression survived and they express these proteins at the same low levels as tumor cell-lines derived from patient tissues. Such transfected clones were used for H3 modification analysis. The reasons for using H4 as loading control are as follows: 1) The ratio between H4 and H3 in nucleosomes/chromatin is fixed and stable. 2) H4 has a slightly lower molecular weight allowing for simultaneous antibody-probing and detection of H3 and H4 without stripping and re-probing western filters or running separate gels. This in turn allows for much more accurate quantification of H3 modifications. 3) H3 is modified at more sites and with more alternative modifications than H4 and, perhaps as a result of this, we were not able to find a reliable pan-H3 antibody for western blot to use as a reference. Taken together, in our hands H4 was a much more reliable loading control/reference than H3.

#5 EGFP toxic effects such as ROS production (for review see Ansari et al., PMID:27435468) might have been the reason for the observed moderate signal shift.

Note that our GFP- only transfected control cells (New added data) express at least 50-100 times more GFP than the tagged oncoprotein transfected cells. We are aware that these high levels of GFP causes ROS stress and have indeed observed such effects when working with an unrelated project (Jauhiainen et al 2012). However, even these extremely high levels of GFP caused no increased H3K27 methylation as was seen in the FET-oncogene transfected cells. Furthermore, the very low GFP levels in the fusion-oncogene transfected cells would generate much less, if any, ROS stress and we have seen no expression of genes known to be induced under ROS stress conditions (affymetric micro array results Jauhiainen 2012 and Engstrom et al 2005). The histone methylation effect in the fusion oncogene transfected cells are therefore not an effect of ROS-stress but instead a result of the FET oncoprotein activities. See response on this item above.

#6 As previous reports did not observe changes in the repressive H3K27me3 mark upon knockdown of EWS-FLI1 in Ewing sarcoma cell lines, but instead a strong decrease in the activating H3K27ac mark, clarification of the functional consequences of FET fusion oncogene interaction with BAF on enhancer activity requires a thorough investigation of this issue. Consequently, the authors should also test for H3K27ac levels in their HT1080 model system.

Again, thanks for constructive suggestion. We have made the requested acetylation analysis, both on the new analysis of stable cell lines and the “old” transient transfection samples. The results are, as expected very variable since acetylation and deacetylation is a highly dynamic and rapidly changing process. Our results show no general trends in increased H3K27 acetylation. These results and discussion on them are added to the manuscript. We think the main reason for not detecting increased acetylation may be that the HT1080 cell line which is our experimental system, has a constitutive expression of FLI1. The normal FLI1 expectedly binds the same target sequences as EWSR1-FLI1 and could thus block out the EWSR1-FLI1 effect on acetylation. The reason why increased K27 trimethylation was not detected in the Boulay paper, could be their use of ChIPseq data which is excellent for identification of binding sites but may, due to normalization of data fail to see minor but globally spread methylation changes. (compare with micro array or RNAseq experiments where normalization of data leads to failure in detection of globally elevated or decreased transcription). In any case, our work, using a different experimental setup contributes with data not observed by Boulay et al. Future studies by us and others will show which data can be confirmed and what the crucial oncogenic mechanisms are. In this context it is also interesting to note that the DDIT3 transcription factor partner of the FUS-DDIT3 fusion protein was originally suggested to be a dominant negative dimerization partner of C/EBP family TFs since the acidic DNA binding domain of DDIT3 was expected to block DNA binding. Later, stress induced co-expression of DDIT3 and ATF4 and heterodimers formed between these proteins was reported to bind DNA and activate transcription. In MLS the normal DDIT3 and ATF4 are silent whereas the FUS-DDIT3 expression is driven by the FUS-promoter. DDIT3-ATF4 dimers cannot form and DNA binding is very limited. (Recent ChIPseq data:

ChIP atlas

SRX359992	SRA104261	GSM1239560	hg19	LoVo
SRX119350	SRA050029	GSM873424	mm9	MEF
SRX119351	SRA050029	GSM873425	mm9	MEF

and our own preliminary results) show DDIT3 binding at very few loci in the genomes of MLS lines unless it is co-expressed and form dimers with the ER-stress induced ATF4. This supports the view that DDIT3, under non-stress conditions, mainly is acting as a dominant negative with little DNA binding. The oncogenic effect of FUS-DDIT3 is therefore probably not as much dependent on massive DNA binding or relocation of SWI/SNF to specific target sequences/sites. This shows that other as yet unknown mechanisms, reflected by an increased H3K27 trimethylation and muting of genes, could be the oncogenic mechanism. As we see an increased H3K27 trimethylation also with EWSR1-FLI1 expression, this could

also be a general oncogenic mechanism for all the FET oncoproteins! The discussion text on this item has been edited in the manuscript.

Referee #2:

The study by Thomsen et al describes the physical interaction between the FET family of transcription factors and members of the SWI/SNF remodeling complex.

The FET family of proteins consists of FUS, EWSR1 and TAF15 and are normally expressed in all tissues and engage at different stages of gene expression. The human FET proteins are associated with transcription, splicing, microRNA processing, RNA transport, signaling and maintenance of genomic integrity. Fusion proteins containing the N-terminal domain of FETs and the C-terminal domain of DNA binding proteins are commonly found in many sarcomas and have been shown to be the driving force for carcinogenesis in these types of cancer.

The mammalian SWI/SNF complex is an ATP-dependent chromatin remodeler composed of 12-15 core subunits that regulate genomic architecture and DNA accessibility. Recent studies revealed that genes encoding the different SWI/SNF subunits are collectively mutated in over 20% of human cancers and that specific subunits are mutated in different cancer types suggesting tissue specific functions.

Using GST-tagged recombinant constructs as baits in pulldown experiments with sarcoma cell line extracts and mass spectrometry, the authors demonstrate that the N-terminal domain of all 3 FETs can strongly interact with members of the mammalian SWI/SNF complex. Moreover, the authors use a proximity ligation assay approach to confirm that both the wild type FUS and the fusion FUS-DDIT3 proteins interact with SWI/SNF components ARID1A and BRG1 in vivo, in the same sarcoma cell lines used for Mass Spec. By generating several truncated GST-tagged constructs of the N-terminal domain of FETs, they specifically identify the area of the FETs responsible for the interaction with SWI/SNF. Furthermore, the authors perform a number of co-IP experiments for BRG1 ("reverse" IP) to confirm the interaction with wild type or fusion FETs and differential salt extraction to demonstrate the strength of the interaction for each SWI/SNF component identified. Finally, in order to get mechanistic insight into the effect that these FET-SWI/SNF interactions have on SWI/SNF function, the authors perform immunoblot analysis for the H3K27me3 modification in cell lines expressing either wild type or fusion FETs and find that the levels of this modifications are elevated in the fusion expressing cells. Given that SWI/SNF antagonizes the polycomb repressive complex 2 (PRC2) which catalyzes the H3K27me3 mark, the authors suggest that the H3K27me3 levels correlate with perturbed SWI/SNF function.

From the present analysis, the authors propose 2 major findings, the first that the N-terminal domain of all 3 FET proteins interacts with many members of the SWI/SNF complex and the second, that this interaction disrupts the normal role of SWI/SNF to oppose the PRC2 suppressive complex and thus this interaction leads to increased EZH2 enzymatic activity and increased H3K27me3 levels.

Given the importance of the FET fusion onco-proteins as driving events in sarcomas, the poor prognosis and difficulty of treatment of these cancers and the importance of the SWI/SNF complex for transcription, the identification of the FET-SWI/SNF interaction is of high importance.

The manuscript is well written and, at least for Figures 1-3, the data is fairly robust and supportive of the conclusions presented. However, the data presented in Figure 4 and Supplementary Figure 3 are weak and correlative. In this reviewer's opinion, while the experiments are generally well performed, the novelty is minimal and the functional insight very weak and correlative (Figure 4), reducing enthusiasm for publishing the paper in EMBO reports.

Our results provide a new unifying pathogenic mechanism for a large group of tumors, all caused by FET oncogenes. This alone makes it an important finding that catches interest from many scientist and clinicians working with these tumors and in addition, scientists working with the SWI/SNF chromatin remodelers. Our data concerning binding profiles between FET and SWI/SNF proteins are completely novel findings and provides a platform for further detailed investigations how FET proteins bind and act on the SWI/SNF complex. We are convinced that this manuscript will be met with interest from a wide group of readers!

More specifically, the major concerns are:

1. Lack of novelty.

A recently published study (Boulay *et al.* *Cell* 171, 2017) demonstrated in detail how the Ewing sarcoma fusion protein EWS-FLI1 interacts with many members of the SWI/SNF complex and provided functional insight for the altered targeting of SWI/SNF in this setting. In the present study, the authors conduct a more thorough analysis in terms of the specific domains of FET proteins that interact with SWI/SNF members and demonstrate the interaction in different types of sarcomas other than Ewing but nevertheless, there is significant overlap with the Boulay *et al* 2017 paper.

We agree that there are some overlap with the excellent Boulay paper. However, Boulay *et al* specifically investigated one of the many FET oncogenes whereas we have focused this work on the common N-terminal parts shared by all the FET oncogenes. Also, we show that two FET-oncoproteins that are very different with regard to their DNA binding profiles, show similar binding to SWI/SNF and similar downstream effects (See also response on this point to the previous referee). Our focus was the protein-protein interaction and only second to this, the downstream effects. Again, our results provide a working mechanistic model for all the FET oncogenes.

2. Weak functional analysis

Although one of the major roles of SWI/SNF is to oppose the function of PRC2, the levels of H3K27me3, especially with the limitations of detection of immunoblot, are not the best readout of SWI/SNF function. A number of recent reports have demonstrated that SWI/SNF binds to enhancer elements and regulate the H3K27ac mark and the accessibility of chromatin for transcription in these areas. In this sense, alteration in global levels of H3K27ac is a better readout of SWI/SNF perturbed function. Furthermore, even if H3K27me3 was a reliable readout of SWI/SNF function, the change in global levels as seen in Figure 4 between HT1080 expressing either wild type FUS or fusion FUS-DDIT3, is marginal and in no way, supports a perturbed SWI/SNF function.

Increased H3K27me3 was considered as a reliable readout in several previous studies of mutated SWI/SNF complexes (See papers referred to in the manuscript). Our initial finding of the increased H3K27me3 increase was not anticipated by us but after many repeated experiments (as a response to the referee comments, we now also added data with stably FUS-DDIT3 and EWSR1-FLI1 transfected cells, (see results part and extended view material) we conclude that H3K27me3 was elevated in every single one of these experiments regardless if they were made with stable or transient transfections. We have spent a lot of time and efforts to fine tune and critically review the semi-quantitative western blot analysis (also added data on a technical control with EZH2 inhibited cells), and we are confident that the FET oncogenes causes a minor but robust increase in H3K27me3.

Also, considering the abundancy of the SWI/SNF complex in human cells and the low expression of the FET oncoproteins in tumor cells and transfected cells (See discussion above with previous reviewer), it is not surprising that only minor fractions of the SWI/SNF complex bind FET oncoproteins and are affected by them. Therefore, a major change in H3K27me3 levels cannot to be expected, but also minor changes at the level reported here, may affect hundreds or even thousands of genes.

On a technical note, the cell lysis and protein extraction method used for histone modification analysis is not the appropriate one as it is known that the majority of histone proteins precipitate in an insoluble fraction. A more specific histone extraction protocol is needed in order to extract the majority of histone proteins from chromatin and have confidence that one has a representative cell extract.

Thanks for this note! As revealed by this referee, our method part of the histone extraction protocol was not clear. We harvested the whole cells in RIPA buffer, sonicated the suspensions without prior centrifugation. Protein levels were then measured and samples from the sonicated suspensions were mixed with reducing agents and LDS sample buffer to a final concentration of more than 2% LDS/SDS. These samples were then heated to 95°C for 10 minutes before loading on the gels. According to my experience, histone proteins are well solubilized under these conditions.

We tested several protocols for histone extraction/analysis but found this to be the most reliable. Note that no material was lost in centrifugation and that the final sample mix was clear, with no signs of precipitates and easy to pipet onto the gel. Details of the protocol description have been edited for clarification of this item in the manuscript.

Referee #3:

In this work Thomsen, Lindén et al. describe the direct physical interaction between the N-terminal part of

the FET protein members EWSR1, FUS and TAF15 and the SWI/SNF chromatin-remodeling complex. Of particular interest, they show how the FUS-DDIT3, EWSR1-FLI1 and EWSR1-ERG translocations display a stronger interaction with the SWI/SNF complex, compared to their wild type counterparts, which lead to a functional change in SWI/SNF activity, increasing the deposition of the H3K27me3 histone modification. The authors suggest that the difference in the interaction strength between the wild-type and translocated FET proteins may underlie, at least in part, the oncogenic role of FUS and EWSR1 N-terminal moieties in Mixoid Liposarcoma and Ewing sarcoma.

The manuscript is well written and the experimental approach, the rationale and the results are clearly presented, and appropriately supported by the experimental data. The topic is very timely and addresses a major question in field on sarcoma pathogenesis, namely, the role of the N-terminal portion of FET proteins in the translocations involved in a wide range of tumors, and the potentially critical functional differences with their wild-type counterparts. Mutations in SWI/SNF protein family members have been associated with more than 20% of human malignancies, and may represent a key oncogenic event in human cancer, offering a new promising Achilles' heel for some of these entities. In translocated human sarcomas, particularly Ewing sarcoma (expressing the EWSR1-FLI1 fusion protein), the role of the EWSR1 N-terminal fragment has been recently suggested to play a critical role in allowing prion-like aggregation of EWSR1-FLI1 with wild type EWSR1 complexes, which are indispensable to activate the oncogenic program sustaining tumor growth. The present work confirms and expands this knowledge to other tumor entities, including Mixoid Liposarcoma (expressing the FUS-DDIT3 translocation), and suggests that a similar mechanism may be active in a broader panel of human sarcomas, driven by chromosomal translocations involving FET proteins.

Although the biochemical characterization of the interaction between FET proteins and the SWI/SNF complex is very detailed and convincing, the reviewer would appreciate some additional evidences of the role of this interaction in tumor development to be added to the current work.

Major points:

1) In Figure 2 the authors show the importance of the 31-66 amino acidic portion of the FUS protein in the interaction with SWI/SNF complex. Along with the difference in binding strength between the wild type and translocated FET proteins, this is a major point of the work. It would be important to add some functional evidences to the role of these mutants in the development of Mixoid Liposarcoma. First of all, the authors should create at least some of these mutants for the FUS-DDIT3 fusion protein, and add the mutant carrying the 31 to 66 amino acidic part of FUS, fused to DDIT3. These mutants could be very useful in determining the importance of the SWI/SNF-FUS interaction for the transforming ability of FUS-DDIT3. Good suggestions! We have tried transformation experiments with FET-oncogenes in different settings for more than 20 years of research but this failed in our hands! Transformation assays with FUS-DDIT3 and EWSR1-FLI1 *in vitro* and *in vivo* has given contradictory results also from other research groups. For example, some groups report that FUS-DDIT3 in transgenic mice or in transduced mouse mesenchymal stem cells cause tumors in mice or and/ or transformation in *in vitro* assays. Others report that additional mutations such as P53 Knock outs are needed. Although we are convinced that FUS-DDIT3 is a tumor causing gen that may act, sometimes without assistance from other oncogenes/suppressor genes, in our own experiments with stable FUS-DDIT3 transduced human fibroblasts, mesenchymal stem cells, or embryonal stem cells, we have not been able to see transforming effects *in vitro* or tumor formation in mice. Thus, without an effective transformation assay, we are not able to test effects of mutations in the fusion oncogene.

Second, by expressing these mutants in unrelated cell types (like SW872 or HT1080 cell lines) they could assess if the mutants are still able to bind to the DNA and activate some of the reported target genes (MMP2, MMP9, Pentaxin 3 and Fibronectin 1). Moreover, since the FUS-DDIT3 translocation has been reported to be able to transform NIH 3T3 cells *in vitro* and *in vivo*, the authors could express the wild type FUS-DDIT3 fusion protein along with the mutants and assess the changes in transforming ability (soft agar or tumorigenicity assays in immunosuppressed mice).

Our attempts to replicate 3T3 assays with FUS-DDIT3 have failed to give reliable results and making stable transfections in other cells than HT1080 and in stem cells is problematic since surviving cells goes senescent. HT1080 avoids senescence, likely since the P16 gene is lost in these cells. These items are also outside the focus of this study.

2) In Figure 4, the authors claim that the expression of the FUS-DDIT3 and EWSR1-FLI1 proteins in HT1080 cells result in an increase in the deposition of the H3K27me3 repressive histone mark. These data

are in conflict with two recent reports describing changes in H3K27ac mark but not H3K27me3 (as noted by the authors). Since the induction of H3K27me3 deposition is relatively subtle, and the timing of the H3K27me3 changes extremely rapid (24 to 48 hours after transfection, which is not a classical time frame for these type of chromatin remodeling events), the reviewer feels that a confirmation of these results using a different approach could be important.

Our initial finding of the increased H3K27me3 increase was not anticipated but after many repeated experiments (as a response to the referee comments, we now also added data with stably FUS-DDIT3 and EWSR1-FLI1 transfected cells, see results part and extended view material), we conclude that H3K27me3 was elevated in every single one of these experiments regardless if they were made with stable or transient transfections. We have spent a lot of work and efforts to fine tune and critically review the semi-quantitative western blot analysis (also added data a technical control with EZH2 inhibited cells), and we are confident that the FET oncogenes causes a minor but robust increase in H3K27me3.

Also, considering the abundancy of the SWI/SNF complex in human cells and the low expression of the FET oncoproteins in tumor cells and transfected cells (See discussion above with previous reviewer), it is not surprising that only minor fractions of the SWI/SNF complex bind FET oncoproteins and are affected by them. Therefore, a major change in H3K27me3 levels can not to be expected. Also minor changes at the levels reported here, may affect hundreds or even thousands of genes.

One possibility would be a detailed Mass Spectrometry analysis of histone H3 modifications, with a quantitative determination of the levels of H3K27me3 in wild type versus FUS-DDIT3 ad EWSR1-FLI1-expressing cells. In addition, since the EWSR1-FLI1 protein has been shown to induce the transcription of EZH2, one of the core members of the PRC2 complex involved in H3K27me3 deposition, the authors should consider the possibility of an indirect effect, not mediated by a competition with the SWI/SNF complex, but by an increase of the PRC2 activity. To this end, the authors should verify if in their cellular model if the expression of FUS-DDIT3 and EWSR1-FLI1 results in the increase of the PRC2 protein members EZH2, EED and SUZ12. This should be performed on a longer time point using a stable lentiviral approach.

Thanks for valuable suggestions. We followed this advice, analyzed EZH2-expression, the methyltransferase enzyme of PRC2, both in transient transfected samples and in the histone modification analysis of stable cell lines, and added the data to the manuscript, (Figure 4A and EV2). As reported by other groups, silencing mutations in some SWI/SNF components lead to increased EZH2 expression and H3K27trimethylation. These are similar to the findings in FET oncogene expressing cells. At this time, we have not access to facilities necessary for quantitative mass spec analysis of histones.

Minor points:

1) In Figure 3A the size of the EWSR1-FLI1 protein seems to be different between the first and last lanes. As pointed out by this referee, the figure orientation was not made optimal here. We have corrected this in the updated manuscript. The original western blots can be checked in the appendix files!

2nd Editorial Decision

28 June 2018

Thank you for the submission of your revised manuscript to our editorial offices. We have now received the reports from the referees that were asked to re-evaluate your study (you will find enclosed below).

As you will see, all referees #2 and #3 now support the publication of your manuscript in EMBO reports. However, referee #1 states that further revision would be necessary. In particular he feels that the following further experimental data need to be added:

- pull down of wildtype FET proteins in absence and presence of overexpressed fusion proteins, and testing for potential changes (reduction) in the amount of free BAF complex.
- testing the deletion of the FET binding domain for BAF interaction in the context of full length FET wildtype and oncogenic fusion proteins.
- knockdown of the fusion FET protein in at least one of the two tumour types, MLS or Ewings, to confirm an effect on histone methylation.

We feel that the manuscript would be significantly strengthened when these data are added, and would therefore encourage you to do so in a final revised version.

Further, I have these editorial requests we also ask you to address:

- Regarding data quantification and statistics, please specify in Figs. 3B/D, 4B, EV2E-J and Appendix Fig. S1 the number "n" for how many independent experiments (biological replicates) were performed, and add error bars (e.g. SEM, SD), and provide statistical testing where applicable, and also provide information regarding the test used to calculate p-values in the respective figure legends. Please also add a paragraph to the Methods section explaining the statistical analyses used throughout the paper. See: <http://embor.embopress.org/authorguide#statisticalanalysis>

- Could the the mass spec. raw data be deposited at a public database (Pride, PeptideAtlas)? See also: <http://embor.embopress.org/authorguide#datadeposition>

- Thanks a lot for providing the source data. Please provide the source data files for the main figures combined, one pdf file per figure.

- a Microsoft Word file (.doc) of the revised manuscript text
- a point-by-point response detailing your responses to the referee comments in Word format (.doc)
- editable TIFF or EPS-formatted figure files (main figures and EV figures) in high resolution (of those with adjusted panels or labels).

In addition I would need from you:

- a short, two-sentence summary of the manuscript
- two to three bullet points highlighting the key findings of your study
- a schematic summary figure (in jpeg or tiff format with the exact width of 550 pixels and a height of about 400 pixels) that can be used as a visual synopsis on our website.

REFeree REPORTS

Referee #1:

This manuscript addresses two major aspects of interaction between FET oncogenes and the BAF chromatin remodeling complex: i) What is shared/different between the FET oncoproteins and the wildtype FET proteins with respect to the BAF complex interaction, and ii) what are the functional consequences. In response to reviewers' comments the authors added new data to their study obtained by ectopically expressing FET fusion proteins in HT1080 cells. They find that forcing FET oncoprotein expression does not alter the amount of FET wildtype proteins associated with the BAF complex and conclude that despite different binding affinities there is no competition between oncogenic and normal FET proteins for BAF complex binding. To explain this finding they provide four alternative models and conclude that either wildtype or fusion FET proteins bind to different sites or different subpopulations of the BAF complex. The latter possibility may be favored if there was mutual exclusivity for binding of wildtype or oncogenic FET proteins, and only a fraction of the BAF complex was occupied by wildtype FET proteins. This may be tested by pulling down wildtype FET proteins in absence and presence of overexpressed fusion proteins, and testing for potential changes (reduction) in the amount of free BAF complex. This model would not require different compositions of BAF complex as suggested by model III.

Using deletion constructs of GST fused FET N-terminal domains the authors delineated the minimal BAF binding region of these truncated constructs to a 26 amino acid „FET binding domain". As model IV proposes binding of wildtype and oncogenic fusion proteins to different sites of the BAF complex, the question arises, for which interaction the FET-binding domain is required. Thus, the deletion of the FET binding domain should be tested for BAF interaction in the context of full length FET wildtype and oncogenic fusion proteins. At present, the data do not allow drawing conclusions about the mechanism of BAF perturbation by FET fusion proteins.

The answer to the question of functional consequences of FET oncoprotein binding to the BAF complex on

chromatin also remains an open one. The authors added results from stably and transiently FET fusion protein expressing HT1080 cells to demonstrate a very weak but consistent increase in H3K27me3 in this system. For EWS-FLI1, they demonstrate additionally a decrease in H3K27ac which is in sharp contrast to results obtained by knockdown of EWS-FLI1 in Ewing sarcoma cells in two previous studies that failed to detect any changes in H3K27me3 but demonstrated a clear increase in overall H3K27ac (Riggi et al., 2014; Tomazou et al., 2015). The authors argue that this discrepancy may be due to FLI1 expression in HT1080 cells which may render EWS-FLI1 target genes already highly acetylated before EWS-FLI1 comes in. However, this finding raises the question, if HT1080 is a relevant model for MLS and Ewing sarcoma. Testing their hypothesis directly upon knockdown of the fusion FET protein in at least one of the two tumor types, MLS or Ewings, will be required to confirm an effect on histone methylation. Finally, the authors report a slight increase in EZH2 expression upon ectopically forcing FET fusion protein expression in HT1080 cells and conclude that this may be the reason for increased H3K27me3. While this may be a reasonable conclusion, it is unrelated to binding of FET oncoproteins to the BAF complex as long as no causative role of the FET/BAF complex interaction for EZH2 regulation is reported. Consequently, there is no functional proof for „dysregulation of SWI/SNF providing a unifying pathogenic mechanism for the large group of tumors caused by FET fusion oncoproteins", as the Abstract says.

Minor point: Figure 2 does not allow for any conclusions about stoichiometry of FET/BAF complex binding. The corresponding statement should be removed from the Results section.

Referee #2:

The manuscript by Thomsen et al has been significantly revised and is now suitable for publication in EMBO reports.

The study still suffers from some weak/correlative results regarding the effects of the fusion proteins on EZH2 and SWI/SNF function but overall, the study is of high interest and should be available to the scientific community.

Referee #3:

The authors have convincingly replied to all the concerns the reviewer has raised in the initial round of reviewing. No additional issues need to be addressed.

2nd Revision - authors' response

31 July 2018

EMBOR-2018-45766V3

Point by point response letter to referee comments.

Referee #1

This manuscript addresses two major aspects of interaction between FET oncogenes and the BAF chromatin remodeling complex: i) What is shared/different between the FET oncoproteins and the wildtype FET proteins with respect to the BAF complex interaction, and ii) what are the functional consequences.

The main aim and results of this work was to identify proteins that bind all the three FET NTDs. As the three FET NTDs all carry the oncogenic activity and can replace each other as fusion partners in the FET oncoproteins. Our hypothesis was that all three FET NTDs interact with the same critical partner. Indeed, this turned out to be the case and from our results we conclude that the SWI/SNF is a main common binding partners for all three FET NTDs. This also suggested a common mechanism of action for the three FET-NTDs and corresponding fusion oncoproteins. We have edited the introduction section to clarify background thinking and aims. We then further investigated the nature of the binding to SWI/SNF, what SWI/SNF components were in the fusion oncoprotein complexes, binding quality etc. The functional part

was never considered to be a main part of the manuscript.

In response to reviewers' comments the authors added new data to their study obtained by ectopically expressing FET fusion proteins in HT1080 cells. They find that forcing FET oncoprotein expression does not alter the amount of FET wildtype proteins associated with the BAF complex and conclude that despite different binding affinities there is no competition between oncogenic and normal FET proteins for BAF complex binding. To explain this finding they provide four alternative models and conclude that either wildtype or fusion FET proteins bind to different sites or different subpopulations of the BAF complex. The latter possibility may be favored if there was mutual exclusivity for binding of wildtype or oncogenic FET proteins, and only a fraction of the BAF complex was occupied by wildtype FET proteins. This may be tested by pulling down wildtype FET proteins in absence and presence of overexpressed fusion proteins, and testing for potential changes (reduction) in the amount of free BAF complex. This model would not require different compositions of BAF complex as suggested by model III.

Thanks for interesting thoughts on the binding models.

The question whether BAF complexes simultaneously bind both normal and oncogenic FET proteins is hard to probe. Our MS analysis data of IP samples from tumor cell lines with DDIT3 and FLI1 antibodies show co-precipitation with both SWI/SNF components and normal FET proteins. However, we know from our published data (Thomsen et al 2013), that the oncogenic FET proteins form complexes with their normal variants making it impossible to separate if the normal FET proteins comes together with the FET-oncoprotein-SWI/SNF precipitates or if they comes together with the small fractions of non-bound oncoproteins. Our attempts to IP the normal FET proteins in tumor cells (requires antibodies that are specific for central or C-terminal parts not present in the fusion proteins) have with several tested antibodies not provided any conclusive results.

Normal FET proteins bind each other in homo and hetero-complexes. The sizes of most of these combination fractions, containing normal and oncogenic FET proteins, have so far eluded measurements. With all these unknown parameters, we cannot find a more defined binding model. Note also that IP with DDIT3 and FLI1 antibodies co-precipitate all the variant SWI/SNF components present in the cells suggesting that the oncoproteins can bind many variants of SWI/SNF (Table 1). We have edited the results/discussion text to clarify the models and reasoning behind them. The main changes are found in the last paragraph of page 5 and the first paragraph of page 6 in the manuscript.

Using deletion constructs of GST fused FET N-terminal domains the authors delineated the minimal BAF binding region of these truncated constructs to a 26 amino acid „FET binding domain“. As model IV proposes binding of wildtype and oncogenic fusion proteins to different sites of the BAF complex, the question arises, for which interaction the FET-binding domain is required. Thus, the deletion of the FET binding domain should be tested for BAF interaction in the context of full length FET wildtype and oncogenic fusion proteins.

This project started as an attempt to identify binding partners to the FET part of the FET family of oncogenes. For many of the FET oncogenes there are multiple variant translocations combining smaller or larger parts of the FET proteins with their translocation factor partners. For this study we selected the variants containing the smallest FET parts known to confer the oncogenic activity. Among FUS, EWSR1 and TAF15 fusions, the FUS parts are the smallest whereas the EWSR1 parts are the longest. However, they can replace each other as fusion partners, indicating that regardless of size, they carry the same oncogenic functions. With our FUS deletion mutants we further narrowed down and identified the smallest parts necessary for binding SWI/SNF. The same parts of FUS are also necessary for normal FET protein complex formation and for formation of FUS-DDIT3 nuclear bodies (Goransson, 2009). The FET binding motifs most likely need additional sequences to bind partner proteins, but this has been very difficult to further investigate due to the repetitive sequences building up most of the FET-NTDs.

As the referee comment on, it would be of great interest to further analyze the binding of the fusion proteins. Here we have added data from IP-experiments to test if the transcription factor partner in FUS-DDIT3 by itself could interact with SWI/SNF (In the previously published paper, Boulay et al, the normal transcription factor FLI1 was reported to itself bind SWI/SNF.) The new data are added to the manuscript (Extended view figure 3) and show that very small fractions of normal DDIT3 interact with the SWI/SNF complex. We conclude that the FET-NTD is the most important part for the fusion oncoprotein binding.

The answer to the question of functional consequences of FET oncoprotein binding to the BAF complex on chromatin also remains an open one. The authors added results from stably and transiently FET fusion

protein expressing HT1080 cells to demonstrate a very weak but consistent increase in H3K27me3 in this system. For EWS-FLI1, they demonstrate additionally a decrease in H3K27ac which is in sharp contrast to results obtained by knockdown of EWS-FLI1 in Ewing sarcoma cells in two previous studies that failed to detect any changes in H3K27me3 but demonstrated a clear increase in overall H3K27ac (Riggi et al., 2014; Tomazou et al., 2015). The authors argue that this discrepancy may be due to FLI1 expression in HT1080 cells which may render EWS-FLI1 target genes already highly acetylated before EWS-FLI1 comes in.

However, this finding raises the question, if HT1080 is a relevant model for MLS and Ewing sarcoma.

We agree that the HT1080 cells may not be as good model system as MLS and Ewing sarcoma tumor cell lines. But they are genetically well defined and lack any mutations in the SWI/SNF component genes and the observed effects are still a strong indication that the FET oncogenes affect SWI/SNF-PRC2 interactions. Furthermore, existing MLS lines are all made with SV40 transfections or carry a P53 mutation with associated genetic instability. The Ewing cell lines used for the studies cited above and the Boulay study, were derived from advanced disease patients treated with multiple rounds of mutagenic chemo therapy and consequently carry multiple secondary mutations in cancer associated genes (Check COSMIC data for those cell lines!!).

Testing their hypothesis directly upon knockdown of the fusion FET protein in at least one of the two tumor types, MLS or Ewings, will be required to confirm an effect on histone methylation.

In previous other projects we have repeatedly tried various knock down methods in MLS and EWS cells but found the results unreliable, mainly due to massive off target effects. (The normal FET proteins bind also small RNAs and we hypothesize that the strong off target effects may be a consequence of this feature). Therefore we avoided these methods for this project. Knock down experiments in other cell lines, and other target genes however, work for us.

Finally, the authors report a slight increase in EZH2 expression upon ectopically forcing FET fusion protein expression in HT1080 cells and conclude that this may be the reason for increased H3K27me3. While this may be a reasonable conclusion, it is unrelated to binding of FET oncoproteins to the BAF complex as long as no causative role of the FET/BAF complex interaction for EZH2 regulation is reported. Consequently, there is no functional proof for „dysregulation of SWI/SNF providing a unifying pathogenic mechanism for the large group of tumors caused by FET fusion oncoproteins“, as the Abstract says.

Increased levels of PRC2 complexes and H3K27me3 have previously been reported as a consequence of mutated/lost SWI/SNF components in other cell types (Kadoch et al, Biochemistry 2018, Kadoch et al, Nature Genetics 2017). Previously, we noted that a small amount of the PRC2 component EZH2 co-precipitated with SWI/SNF (BRG1 antibody). Hence, we set up additional IP experiments to check if the interaction between SWI/SNF and PRC2/EZH2 is altered in FET-oncogene expressing cells. Indeed, our results show that overexpression of FUS-DDIT3 increases the fraction of EZH2 that co-precipitates with SWI/SNF. We have thus observed another direct effect on SWI/SNF interaction with PRC2. The new data are added to FIGURE 3 in the manuscript. *Minor point: Figure 2 does not allow for any conclusions about stoichiometry of FET/BAF complex binding. The corresponding statement should be removed from the Results section.*

We agree with the referee and have edited this text section.

Referee #2

The manuscript by Thomsen et al has been significantly revised and is now suitable for publication in EMBO reports. The study still suffers from some weak/correlative results regarding the effects of the fusion proteins on EZH2 and SWI/SNF function but overall, the study is of high interest and should be available to the scientific community.

Referee #3

The authors have convincingly replied to all the concerns the reviewer has raised in the initial round of reviewing. No additional issues need to be addressed.

3rd Editorial Decision

22 August 2018

Thank you for the submission of your revised manuscript to our editorial offices. We have now received the report from the referee that was asked to re-evaluate your study (you will find enclosed below).

As you will see, the referee still has concerns, and asks to address these experimentally. I think these are important points, and the suggested remaining experiments are feasible, and within the scope of the paper. Thus, we ask you to address these experimentally in a further revised manuscript.

Moreover, I think it is not ideal that presently the reader gets the impression that many conclusions of the paper are drawn from experiments performed once. It will strengthen the message significantly, if the quantifications shown in Figs. 3B, 3D, EV2F-J, and EV3 B are provided from at least three independent experiments, including proper statistics, and statistical testing. See also:
<http://embor.embopress.org/authorguide#statisticalanalysis>

Thus, please add replicates and statistics, and update the respective graphs.

We plan to publish this paper as Scientific Report. For a short report, you could use up to 5 main figures and up to 5 EV figures. Thus there is plenty of space for additional data, and an Appendix would not be necessary. For more details please refer to our guide to authors:
<http://embor.embopress.org/authorguide#manuscriptpreparation>

Further, I have these remaining editorial requests:

- Please deposit the mass spec. raw data a public database (Pride, PeptideAtlas)? See:
<http://embor.embopress.org/authorguide#datadeposition>
- Please include all materials and methods in the main manuscript file.
- There are inconsistencies in the author contributions. It seems Christer Thomsen is currently mentioned as CL, and Pernilla Grundevik as PL. Please check this, and correct.
- Please have the manuscript proof-read by a native speaker (see also the minor point of referee #1).

- a Microsoft Word file (.doc) of the revised manuscript text
- a point-by-point response detailing your responses to the referee comments in Word format (.doc)
- editable TIFF or EPS-formatted figure files (main figures and EV figures) in high resolution (of those with adjusted panels or labels).

In addition I would need from you:

- a short, two-sentence summary of the manuscript
- two to three bullet points highlighting the key findings of your study
- a schematic summary figure (in jpeg or tiff format with the exact width of 550 pixels and a height of about 400 pixels) that can be used as a visual synopsis on our website. This should be a summary sketch, highlighting the major findings of the paper.

REFEREE REPORT

Referee #1:

The authors make a strong point that the goal of the study was the discovery of proteins that interact with all the FET fusion proteins, but not to interrogate the functional consequences. Yet, already the title and the Abstract point out a functional perturbation of the SWI/SNF complex by binding to FET fusion proteins, which needs to be undermined by experimental proof. The authors added new data to Figure 3 of their manuscript to demonstrate an increase in EZH2 protein association with SWI/SNF upon FET fusion protein expression. However, the representative immunoblot of the IP experiment is not at all convincing to allow for this conclusion.

The question if FET wildtype and FET oncogenic fusion proteins bind simultaneously to the same or distinct SWI/SNF complexes is an important one, as the authors demonstrate higher affinity of the fusion proteins to SWI/SNF than of the wildtype proteins. Yet, it still remains unsolved. The authors refer to the promiscuity of wildtype FET proteins and the difficulty to perform pulldowns with antibodies to their C-terminus. These points are well taken. Still the authors could have performed SWI/SNF pull down assays from cells transfected with FET and FET-oncogene fusion constructs with a deletion of the 26 amino acid FET binding domain to ask if both interact with SWI/SNF through the same protein interaction surface. In the revised manuscript, the authors have added IP experiments testing the interaction of wildtype DDIT3 with SWI/SNF and reporting only weak interaction. They therefore conclude that it is the FET-NTD that has to be responsible for the strong co-IP with the FET-DDIT3 fusion protein. This, however, is only indirect evidence. As reported earlier and referenced by the authors, the NTD may lead to aggregation of the fusion protein and interaction with SWI/SNF may still take place via the DDIT3 domain or a composite interaction surface of the fusion protein. In case that, different from the wildtype FET proteins, FET-fusion proteins with a deletion of the FET interaction domain still co-IP with SWI/SNF, it is likely that wildtype and fusion proteins occupy different sites of the SWI/SNF complex through different domains. Also, combining constructs (wildtype or fusion protein) with the 26 amino acid deletion with those without in mutual combinations may further help to rule out (or not) model 2.

In response to this reviewer's critique that results obtained by ectopic FET fusion protein expression in HT1080 sarcoma cells may not be representative of MLS and Ewing sarcoma cells with regards to the observed epigenetic effects, the authors question the relevance of available MLS and Ewing sarcoma cell lines. Specifically, they argue that the Ewing sarcoma cell lines, which have been used in previous epigenomic studies to demonstrate enhanced EWS-FLI1 driven H3K27 acetylation, were derived from advanced disease and harbor a number of additional mutations. They imply that the proposed increase in H3K27 tri-methylation observed in HT1080 cells upon ectopic FET fusion protein expression may be converted to the opposite effect in MLS and Ewing sarcoma cell lines by additional mutations such as of p53. This is pure speculation. Even though the authors may not be able to knockdown FUS-DDIT3 in MLS cell lines, knockdown of EWS-FLI1 in Ewing cells to test their hypothesis on H3Kme3 is well established and could be performed in a p53 wildtype cell line.

Minor: there are a few grammar mistakes in the manuscript that should be corrected.

3rd Revision - authors' response

15 February 2019

Referee #1:

A: The authors make a strong point that the goal of the study was the discovery of proteins that interact with all the FET fusion proteins, but not to interrogate the functional consequences. Yet, already the title and the Abstract point out a functional perturbation of the SWI/SNF complex by binding to FET fusion proteins, which needs to be undermined by experimental proof. The authors added new data to Figure 3 of their manuscript to demonstrate an increase in EZH2 protein association with SWI/SNF upon FET fusion protein expression. However, the representative immunoblot of the IP experiment is not at all convincing to allow for this conclusion.

After repeating the experiment, we agree that the change in the EZH2 protein interaction with SWI/SNF after forced expression of FET fusions is not statistically significant and have changed the text accordingly. Our data, however, show after repeated experiments with biological replicates that forced expression of FET oncogenes *FUS-DDIT3* and *EWSR1-FLI1* caused increased H3K27me3. Furthermore, new RNA-seq-based experiments show that *FUS-DDIT3*-regulated genes overlap significantly with genes regulated in PRC2 or BAF57/SMARCE1 (a SWI/SNF component) knockdown cells (new RNA-seq data with

bioinformatics). The data support our hypothesis that deregulation of this chromatin remodeling complex is most likely a central oncogenic mechanism for FET oncogenes.

B. *The question if FET wildtype and FET oncogenic fusion proteins bind simultaneously to the same or distinct SWI/SNF complexes is an important one, as the authors demonstrate higher affinity of the fusion proteins to SWI/SNF than of the wildtype proteins. Yet, it still remains unsolved. The authors refer to the promiscuity of wildtype FET proteins and the difficulty to perform pull-downs with antibodies to their C-terminus. These points are well taken. Still the authors could have performed SWI/SNF pull down assays from cells transfected with FET and FET-oncogene fusion constructs with a deletion of the 26 amino acid FET binding domain to ask if both interact with SWI/SNF through the same protein interaction surface.*

We agree that the question if normal and oncogenic FET proteins bind SWI/SNF with the same protein interaction surface would be interesting to study further. However, in several separate projects aiming at structure /interaction analysis etc., we have previously produced and tried to analyze various recombinant variants of the FET-NTDs *in vivo* and *in vitro* transcription/translation systems. The frustrating lessons from those projects are that removing parts from these prion-like domains causes dramatic and unexpected behavior with aggregation, instability, poor expression and variable nuclear to cytoplasmic ratio. Anyway, we have tried to meet the suggestions of the referee and tested additional deletion mutants by transfection into HT1080 cells, followed by IP/western analysis of nuclear extracts (see Figure 1 below in this letter). As we already experienced, the IP results are very variable and the amounts of proteins produced and nuclear/cytoplasmic distribution is extremely divergent between the constructs. We have given up these attempts for now. We do not find the data reliable or useful and have not included them in the manuscript.

A. Comparison bound fraction deletion mutants after BRG1 Co-IP. Immunoblot analysis (IB) of EGFP-tagged deletion mutants of FUS-NTD and FUS-DDIT3 proteins (transiently transfected for 24 h in HT1080 cell line) co-immunoprecipitated with BRG1, detection with antibodies against BRG1 and GFP. Present amino acid numbers are indicated.

B. Quantitative comparison deletion mutants after BRG1 Co-IP. Immunoblot analysis (IB) of EGFP-tagged proteins co-immunoprecipitated with BRG1. Detection with antibodies against BRG1 and EGFP in the HT1080 cell line transiently transfected (24h) with deletion mutants of FUS-NTD and FUS-DDIT3 proteins. Relative amounts of protein, for each sample, were loaded on the gel in order to directly quantify the fraction of bound and non-bound protein. However, due to the difference in expression amount and difference in cytoplasmic (CF) and nuclear (I, input) localization, accurate estimation of the interaction signal based on quantification signals is impossible. B: bound proteins, NB: proteins not bound. Present amino acid numbers are indicated.

C. *In the revised manuscript, the authors have added IP experiments testing the interaction of wildtype DDIT3 with SWI/SNF and reporting only weak interaction. They therefore conclude that it is the FET-NTD that has to be responsible for the strong co-IP with the FET-DDIT3 fusion protein. This, however, is only indirect evidence.*

Our data from GST-tagged FUS-NTD in pull-down experiments provides direct evidence that this domain by itself binds very robustly to the SWI/SNF (Figures 1 and 2 in the manuscript). Most likely the transcription factor partners of the fusion proteins also contribute to the SWI/SNF binding. We have adjusted the manuscript text to clarify this point.

D. *As reported earlier and referenced by the authors, the NTD may lead to aggregation of the fusion protein and interaction with SWI/SNF may still take place via the DDIT3 domain or a composite interaction surface of the fusion protein. In case that, different from the wildtype FET proteins, FET-fusion proteins with a deletion of the FET interaction domain still co-IP with SWI/SNF, it is likely that wildtype and fusion proteins occupy different sites of the SWI/SNF complex through different domains. Also, combining constructs (wildtype or fusion protein) with the 26 amino acid deletion with those without in mutual combinations may further help to rule out (or not) model 2.*

As answered above under **B**, we have tried to further address these questions but failed due to problems caused by the prion-like nature of the FET proteins and natural strong expression of all three interacting FET proteins. We hope to return to these issues when future methodology and experimental systems permit such experiments.

E. In response to this reviewer's critique that results obtained by ectopic FET fusion protein expression in HT1080 sarcoma cells may not be representative of MLS and Ewing sarcoma cells with regards to the observed epigenetic effects, the authors question the relevance of available MLS and Ewing sarcoma cell lines. Specifically, they argue that the Ewing sarcoma cell lines, which have been used in previous epigenomic studies to demonstrate enhanced EWS-FLI1 driven H3K27 acetylation, were derived from advanced disease and harbor a number of additional mutations. They imply that the proposed increase in H3K27 tri-methylation observed in HT1080 cells upon ectopic FET fusion protein expression may be converted to the opposite effect in MLS and Ewing sarcoma cell lines by additional mutations such as of p53. This is pure speculation.

We agree there is no question of the relevance of MLS and EWS sarcoma cell lines; they have an obvious purpose as model systems for the tumors we want to study. Both mutations and epigenetics/chromatin landscape differs between cell lines and will surely affect the outcome of the experiments. We agree with the referee that the results obtained from our experimental model with HT1080 cells probably reflect only a partial and not a complete picture of the changes made by FET oncoproteins in real tumors. For example, no EWSR1-FLI1-induced H3K27ac increase is seen in this model but is convincingly reported in other studies. As mentioned in the manuscript, these results may be partially explained by the high expression of normal FLI1 in HT1080 cells. The FUS-DDIT3 fusion contains the transcription factor partner DDIT3, which is regarded mainly as a suppressing transcription factor. For this FET oncogene, but not for EWSR1-FLI1, the suppressive effects, combined with H3K27 methylation levels are supported by new RNA-seq + bioinformatics data. We have edited the text to clarify and discuss these points.

Our RNA-seq data, where genes up/down-regulated after expression of EWSR1-FLI1 significantly overlapped with several Ewing sarcoma datasets (knockdown/expression of EWSR1-FLI1), supports the use of HT1080 cells with forced EWSR1-FLI1 expression as a EWS model system (Figure 4C and source data).

F. Even though the authors may not be able to knockdown FUS-DDIT3 in MLS cell lines, knockdown of EWS-FLI1 in Ewing cells to test their hypothesis on H3Kme3 is well established and could be performed in a p53 wildtype cell line.

We have again tried our best to knockdown FUS-DDIT3 in an MLS cell line. Partial knockdown was accomplished in some experiments but no reduction in H3K27me3 was observed. Instead, EZH2 levels increased accompanied by H3K27me3 levels (Figure 2 in this letter). We are hesitating how to interpret these results since siRNA treatments to knock down genes are known to often cause more off target effects than on target effects (see for example: Schultz et al doi:10.1186/1758-907X-2-3). In this case, the increased EZH2 levels may indicate an off target effect.

Method: For the (FUS)-DDIT3 knock-down, cells were transfected with either 20 nM DDIT3 siRNA (SI04996488, Qiagen, Hilden, Germany) or 20 nM siRNA control (1027280, Qiagen) for 24h or 48 h using HiPerFect transfection reagent (301707, Qiagen) according to the manufacturer's instructions.

During the revisions, we have substantially strengthened our data regarding histone modifications; we have included information about H3K27Ac, added another control (HT1080 EGFP), analyzed if the increase in H3K27me3 might be explained by an increase in EZH2 (potentially, but not significantly) and added statistical analysis from our four independent replicates. We therefore trust our data that H3K27me3 is increased after forced expression of FUS-DDIT3 (**, p=0.0042) and EWSR1-FLI1 (p=0.0677, if high outlier is removed **, p=0.0010) in HT1080 (see also Figure 4B and source data).

We agree with the referee that more data was needed regarding the functional analysis of FET-oncogene downstream effects. In order to improve this part, we studied the effects of FUS-DDIT3 and EWSR1-FLI1 by RNA-seq and bioinformatic analysis of HT1080 cells transfected with these fusion oncogenes. Lists of up- and downregulated genes were used to search databases of conditionally regulated gene-sets. The results show significant overlaps between FUS-DDIT3 downregulated genes and genes upregulated when PRC2 or SWI/SNF is knocked out or mutated. These results support our model that FUS-DDIT3 affects the SWI/SNF – PRC2 balance. In contrast, the results with FUS-DDIT3 upregulated or EWSR1-FLI1 up- and downregulated genes showed no such overlaps. EWSR1-FLI1-regulated genes showed, however, overlap with gene sets from experiments where EWSR1-FLI expression was manipulated in experimental systems (Figure 4C and source data). The RNA-seq/ bioinformatic results are presented and discussed in a new section of the manuscript.

G. Minor: *there are a few grammar mistakes in the manuscript that should be corrected.*
We have edited the text to improve the language.

4th Editorial Decision

4 March 2019

Thank you for the submission of your revised manuscript to our editorial offices. We have now received the report from the referee that was asked to re-evaluate your study (you will find enclosed below). As you will see, the referee now supports the publication of your study in EMBO reports.

Before we can proceed with acceptance, I have these final editorial requests:

- Please add call-outs for the single panels of the EV figures to the main text. So far, most call outs just refer to Fig. EV1, EV2 or EV3.
- Please edit the legend of Fig. 3D that it becomes clear to the reader why no bar is present in the fusion protein control.
- Please remove the WB data R1 in figure EV2, as this is already shown in Fig. 4A, and refer in the figure legend of Fig. EV2 to Fig. 4A.
- Please change the symbols in Fig. 4B to clearly indicate data from stable and transient transfections.
- Please explain also in the manuscript text what the data in Fig. EV2H-I shows, and how it was obtained (additional data on H3K27me3?).
- If you have replicates for EV2J, please include these.
- In the legend in Fig. EV2G, please remove HT1080 EGFP, as this seems not to be shown in the bar diagrams.

4th Revision - authors' response

6 March 2019

The authors performed all minor editorial changes.

Pierre Aman
EMBO REPORTS
EMBOR-2018-45766